# Emergent Spacetime and Cosmic Inflation

Hyun Seok Yang 

Department of Physics and Photon Science, Gwangju Institute of Science and Technology, Gwangju 61005, Republic of Korea; hsyang@gist.ac.kr

**Abstract:** We present a novel background-independent framework for cosmic inflation, starting with a matrix model. In this framework, inflation is portrayed as a dynamic process responsible for the generation of both space and time. This stands in contrast to conventional inflation, which is characterized as a mere (exponential) expansion of an already existing spacetime, driven by the vacuum energy associated with an inflaton field. We observe that the cosmic inflation is triggered by the condensate of Planck energy into a vacuum and responsible for the dynamical emergence of spacetime. The emergent spacetime picture admits a background-independent formulation so that the inflation is described by a conformal Hamiltonian system which requires neither an inflaton field nor an ad hoc inflation potential. This implies that the emergent spacetime may incapacitate all the rationales to introduce the multiverse hypothesis.

**Keywords:** emergent spacetime; cosmic inflation; quantum gravity

## 1. Introduction

History is a mirror to the future. If we do not learn from the mistakes of history, we are doomed to repeat them.[1] In the middle of the 19th century, Maxwell's equations for electromagnetic phenomena predicted the existence of an absolute speed, $c = 2.998 \times 10^8$ m/s, which apparently contradicted the Galilean relativity, a cornerstone on which the Newtonian model of space and time rested. Since most physicists, by then, had developed deep trust in the Newtonian model, they concluded that Maxwell's equations can only hold in a specific reference frame, called the ether. However, by doing so, they reverted back to the Aristotelian view that Nature specifies an absolute rest frame. It was Einstein who realized the true implication of this quandary: It was asking us to abolish Newton's absolute time as well as absolute space. The ether was removed by Einstein's special relativity by radically modifying the concept of space and time in the Newtonian dynamics. Time lost its absolute standing and the notion of absolute simultaneity was physically untenable. Only the four-dimensional spacetime has an absolute meaning. The new paradigm of spacetime has completely changed the Newtonian world with dramatic consequences.

The physics of the last century was devoted to the study of two pillars: general relativity and quantum field theory, and these two cornerstones of modern physics can be merged into beautiful equations, the so-called Einstein equations given by

$$R_{\mu\nu} - \frac{1}{2}g_{\mu\nu}R = 8\pi G_N T_{\mu\nu}, \tag{1}$$

where the right-hand side is the energy–momentum tensor whose contents are described by (quantum) field theories. Although the groundbreaking theories of relativity and quantum mechanics have utterly changed the way we think about Nature and the Universe, new open problems have emerged which have not been resolved yet within the paradigm of 20th century physics. For example, a short list of these includes the cosmological constant problem, the hierarchy problem, dark energy, dark matter, cosmic inflation and quantum gravity. In particular, recent developments in cosmology, particle physics and string theory

have led to a radical proposal that there could be an ensemble of universes that might be completely disconnected from ours [1]. Certainly, it would be perverse to claim that nothing exists beyond the horizon of our observable universe. The observable universe is one causal patch of a much larger unobservable universe. However, resorting to the concept of the string landscape or multiverse in an attempt to address certain notorious issues in theoretical physics through the anthropic argument is a challenging approach [2]. "And it's pretty unsatisfactory to use the multiverse hypothesis to explain only things we don't understand".[2] Reflecting on history, the current situation strongly echoes the era of the hypothetical luminiferous ether in the late 19th century. The historical lesson implies that we may need another turn of the spacetime picture to defend the integrity of physics.

In physical cosmology, cosmic inflation in the early universe is the exponential expansion of space. Suppose that spacetime evolution is determined by a single scale factor $a(t)$ and its Hubble expansion rate $H \equiv \frac{\dot{a}}{a}$ according to the cosmological principle and driven by the dynamics of a scalar field $\phi$, called the inflaton [3,4]. Then, the Einstein Equation (1) reduces to the Friedmann equation

$$H^2 = \frac{8\pi G_N}{3}\left(\frac{1}{2}\dot{\phi}^2 + V(\phi)\right). \tag{2}$$

The evolution equation of the inflaton in the Friedmann universe is described by

$$\ddot{\phi} + 3H\dot{\phi} + \frac{\delta V}{\delta \phi} = 0. \tag{3}$$

The Friedmann Equation (2) tells us that in the early universe, with $V(\phi) \approx V_0$ and $\dot{\phi} \approx 0$, there was an inflationary epoch of the exponential expansion of space, i.e., $a(t) \propto e^{Ht}$ where $H = \sqrt{\frac{8\pi G_N V_0}{3}}$ is called the inflationary Hubble constant. In order to successfully fit to data, one finds [5]

$$V_0 \geq (2 \times 10^{15}\text{GeV})^4 \approx (10^{-3}M_P)^4 \tag{4}$$

where $M_P = 1/\sqrt{8\pi G_N}$ is the Planck mass.

Let us critically examine the inflationary scenario. According to this scenario [3,4], inflation is described by the exponential expansion of the universe in a supercooled false vacuum state that is a metastable state without any fields or particles but with a large energy density. It should be emphasized that the inflation scenario so far has been formulated in the context of effective field theory coupled to general relativity. Thus, in this scenario, the existence of space and time is a priori assumed from the beginning, although the evolution of spacetime is determined by Equation (1). In other terms, the inflationary scenario does not delineate the generation or creation of spacetime but merely signifies the expansion of preexisting spacetime. It does not delve into the dynamic origin of spacetime. Nevertheless, there has to be a definite beginning so that the quantum gravity era cannot be avoided in the past even if inflation takes place [6]. This implies that the current inflationary scenario is insufficient in describing the initial stage of our universe, and it necessitates the incorporation of new physics to explore the past boundaries of inflating regions.[3] One plausible explanation is the occurrence of a quantum creation as a beginning of the universe [12,13].

The Friedmann Equation (2) reveals that cosmic inflation is triggered by the potential energy associated with an inflaton, whose energy scale is in proximity to the Planck energy. Near the Planck energy, quantum gravity effects become strong and the effective field theory description may break down. If one identifies the slowly varying inflaton field $\phi(t)$ with a particle trajectory $x(t)$ and $\dot{\phi}(t)$ with its velocity $v(t) = \dot{x}(t)$, the evolution Equation (3) tells us that the frictional force, $3Hv(t)$, resulting from the inflating spacetime, is (almost) balanced with an external force $F(x) = -\frac{dV}{dx}$, i.e.,

$$3H\dot{x}(t) \approx F(x), \tag{5}$$

because $\ddot{x} \approx 0$ during inflation. This implies that the cosmic inflation as a dynamical system corresponds to a non-Hamiltonian system.[4]

Recent advancements in string theory have unveiled a remarkable and radical perspective on the nature of gravity. One notable example is the AdS/CFT correspondence, which depicts a surprising scenario wherein $U(N)$ gauge theory in lower dimensions defines a nonperturbative formulation of quantum gravity in higher dimensions [14–16]. In particular, the AdS/CFT duality shows a typical example of emergent gravity and emergent space because gravity in higher dimensions is defined by a gravityless field theory in lower dimensions. Now, numerous examples from string theory illustrate that spacetime is not fundamental but rather emerges only at large distances, constituting a classical approximation [17,18]. Consequently, the governing principle in quantum gravity dictates that space and time are an emergent entity. Since the emergent spacetime, we believe, is a significant new paradigm for quantum gravity, we aim to apply the emergent spacetime picture to cosmic inflation. We will propose a background-independent formulation of the cosmic inflation.[5] This means that we do not assume the prior existence of spacetime but define a spacetime structure as a solution of an underlying background-independent theory such as matrix models. The inflation in this picture corresponds to a dynamical process to generate space and time which is very different from the standard inflation simply describing an (exponential) expansion of a preexisting spacetime. It turns out that spacetime is emergent from the Planck energy condensate in a vacuum that generates an extremely large *Universe*. Our observable patch within the cosmic horizon is a very tiny part $\sim 10^{-26}$ of the entire spacetime. Originally, the multiverse hypothesis was motivated by an attempt to explain the anthropic fine-tuning such as the cosmological constant problem [19] and boosted by the chaotic and eternal inflation scenarios [3,4] and the string landscape derived from the Kaluza–Klein compactification of string theory [20–22], which are all based on the traditional spacetime picture. Since emergent spacetime is radically different from any previous physical theories, all of which describe what happens in a given spacetime, the multiverse picture must be reexamined from the standpoint of emergent spacetime. The cosmic inflation from the emergent spacetime picture will certainly open a new prospect that may cripple all the rationales used to introduce the multiverse hypothesis [23,24].

Given that the concept of the multiverse introduces profound conceptual challenges, compelling us to reconsider the very foundations of science [2], it becomes imperative to carefully contemplate the true nature of the multiverse. Is it merely a speculative illusion stemming from an incomplete physics, akin to the ether in the late 19th century, or does it hold significant relevance even within a more complete theoretical framework? The main purpose of this paper is to illuminate how the emergent spacetime picture brings about radical changes in physics, especially regarding to physical cosmology. In particular, a background-independent theory such as matrix models provides a concrete realization of the idea of emergent spacetime which has a sufficiently elegant and explanatory power to defend the integrity of physics against the multiverse hypothesis [23,24]. The emergent spacetime is a completely new paradigm so that the multiverse debate in physics circles has to seriously take it into account.

This paper is organized as follows. In Section 2, we compactly review the background-independent formulation of emergent gravity and emergent spacetime in terms of matrix models [25–29]. See also closely related works [30–38]. The background-independent formulation of emergent gravity crucially relies on the fact that noncommutative (NC) space arises as a vacuum solution of a large $N$ matrix model in the Coulomb branch and this vacuum on the Coulomb branch admits a separable Hilbert space as quantum mechanics [39]. The gravitational metric is derived from a nontrivial inner automorphism of the NC algebra $\mathcal{A}_\theta$, in which the NC nature is essential to realize the emergent gravity. An important point is that the matrix model does not presuppose any spacetime background on which fundamental processes develop. Rather, the background-independent theory provides a mechanism of spacetime generation such that any spacetime structure including the flat spacetime arises as a solution of the theory itself.

In Section 3, we note that the Planck energy condensate in a vacuum must be a dynamical process. We show that the cosmic inflation arises as a solution of a time-dependent matrix model, describing the dynamical process of the vacuum energy condensation. It turns out that the cosmic inflation corresponds to the dynamical mechanism for the instantaneous condensation of vacuum energy to enormously spread out spacetime. It is remarkable to see that the inflation can be described by time-dependent matrices only without introducing any inflaton field as well as an ad hoc inflation potential. Our work is not the first to address physical cosmology using matrix models. There have been interesting earlier attempts [40–42]. In particular, the cosmic inflation was addressed in very interesting works [43–45] using the Monte Carlo analysis of the type IIB matrix model in Lorentzian signature and it was found that three out of nine spatial directions start to expand at some critical time after which exactly (3 + 1)-dimensions dynamically become macroscopic.

In Section 4, we discuss why the cosmic inflation triggered by the Planck energy condensate into vacuum must be a single event [23,24] and the emergent spacetime precludes the formation of pocket universes appearing in the eternal (or chaotic) inflation. We also discuss a speculative mechanism to end the inflation by some nonlinear damping through interactions between the inflating background and ubiquitous local fluctuations. Finally, we discuss possible ways to understand our real world $\mathbb{R}^{3,1}$ that is unfortunately beyond our current approach because $\mathbb{R}^{3,1}$ does not belong to the family of (almost) symplectic manifolds.

In Appendix A, we briefly review the mathematical foundation of locally conformal symplectic and cosymplectic manifolds that correspond to a natural phase space describing the cosmic inflation of our universe. In Appendix B, we give a brief exposition of a harmonic oscillator with time-dependent mass to illustrate how a nonconservative dynamical system with friction can be formulated by a time-dependent Hamiltonian system, which may be useful to understand the cosmic inflation as a dynamical system. In Appendix C, we propose a background-independent formulation of string theory in terms of matrix string theory [46–49]. We argue that the pseudoholomorphic curve [50] can be generalized to the Hitchin equations describing a Higgs bundle [51,52] by the matrix string theory.

## 2. Emergent Spacetime from Matrix Model

Let us start with a zero-dimensional matrix model with a bunch of $N \times N$ Hermitian matrices, $\{\phi_a \in \mathcal{A}_N | a = 1, \cdots, 2n\}$, whose action is given by [53]

$$S = -\frac{1}{4} \sum_{a,b=1}^{2n} \mathrm{Tr}\, [\phi_a, \phi_b]^2. \tag{6}$$

We require that the matrix algebra $\mathcal{A}_N$ is associative, from which we obtain the Jacobi identity

$$[\phi_a, [\phi_b, \phi_c]] + [\phi_b, [\phi_c, \phi_a]] + [\phi_c, [\phi_a, \phi_b]] = 0. \tag{7}$$

We also assume the action principle, from which we yield the equations of motion:

$$\sum_{b=1}^{2n} [\phi_b, [\phi_a, \phi_b]] = 0. \tag{8}$$

We emphasize that we have not introduced any spacetime structure to define the action (6). It is enough to suppose the matrix algebra $\mathcal{A}_N$ consisted of a bunch of matrices which are subject to a few relationships given by Equations (7) and (8).

First, suppose that the vacuum configuration of $\mathcal{A}_N$ is given by

$$\langle \phi_a \rangle_{\mathrm{vac}} = p_a \in \mathcal{A}_N, \tag{9}$$

which must be a solution of Equations (7) and (8). In particular, we are interested in the matrix algebra $\mathcal{A}_N$ in the limit $N \to \infty$. An obvious solution in the limit $N \to \infty$ is given by the Moyal–Heisenberg algebra[6]

$$[p_a, p_b] = -iB_{ab}, \tag{10}$$

where $(B)_{ab} = -l_s^{-2}(\mathbf{1}_n \otimes i\sigma^2)$ is a $2n \times 2n$ constant symplectic matrix and $l_s$ is a typical length scale set by the vacuum. A general solution will be generated by considering all possible deformations of the Moyal–Heisenberg algebra (10). It is assumed to take the form

$$\phi_a = p_a + \widehat{A}_a \in \mathcal{A}_N, \tag{11}$$

obeying the deformed algebra given by

$$[\phi_a, \phi_b] = -i(B_{ab} - \widehat{F}_{ab}), \tag{12}$$

where

$$\widehat{F}_{ab} = \partial_a \widehat{A}_b - \partial_b \widehat{A}_a - i[\widehat{A}_a, \widehat{A}_b] \in \mathcal{A}_N \tag{13}$$

with the definition $\partial_a \equiv \mathrm{ad}_{p_a} = -i[p_a, \cdot]$. For the general matrix $\phi_a \in \mathcal{A}_N$ to be a solution of Equations (7) and (8), the set of matrices $\widehat{F}_{ab} \in \mathcal{A}_N$, called the field strengths of NC $U(1)$ gauge fields $\widehat{A}_a \in \mathcal{A}_N$, must obey the following equations

$$\widehat{D}_a \widehat{F}_{bc} + \widehat{D}_b \widehat{F}_{ca} + \widehat{D}_c \widehat{F}_{ab} = 0, \tag{14}$$

$$\sum_{b=1}^{2n} \widehat{D}_b \widehat{F}_{ab} = 0, \tag{15}$$

where

$$\widehat{D}_a \widehat{F}_{bc} \equiv \mathrm{ad}_{\phi_a} \widehat{F}_{bc} = -i[\phi_a, \widehat{F}_{bc}] = -[\phi_a, [\phi_b, \phi_c]]. \tag{16}$$

The algebra $\mathcal{A}_N$ admits a large amount of inner automorphism denoted by $\mathrm{Inn}(\mathcal{A}_N)$. Note that any automorphism of the matrix algebra $\mathcal{A}_N$ is inner. Suppose that $\mathcal{A}'_{\widetilde{N}} = \{\phi'_a | a = 1, \cdots, m\}$ is an another matrix algebra composed of $m$ elements of $\widetilde{N} \times \widetilde{N}$ Hermitian matrices. We will identify two matrix algebras, i.e., $\mathcal{A}_N \cong \mathcal{A}'_{\widetilde{N}}$ if $m = 2n$ and $\widetilde{N} = N$ and there exists a unitary matrix $U \in \mathrm{Inn}(\mathcal{A}_N)$ such that $\phi'_a = U\phi_a U^{-1}$, $\forall a = 1, \cdots, 2n$. It is important to note that the NC algebra $\mathcal{A}_N$ generated by the vacuum operators $p_a$ admits an infinite-dimensional separable Hilbert space

$$\mathcal{H} = \{|n\rangle | n = 1, \cdots, N \to \infty\}, \tag{17}$$

that is the Fock space of the Moyal–Heisenberg algebra (10). As is well known from quantum mechanics [55], there is a one-to-one correspondence between the operators in $\mathrm{Hom}(V)$ and the set of $N \times N$ matrices over $\mathbb{C}$ where $V$ is an $N$-dimensional complex vector space. In our case, $V = \mathcal{H}$ is a Hilbert space and $N = \dim(\mathcal{H}) \to \infty$. Thus, the matrix algebra $\mathcal{A}_N$ can be realized as a Hilbert space representation of the NC $\star$-algebra

$$\mathcal{A}_\theta = \{\widehat{\phi}_a(y) \in \mathrm{Hom}(\mathcal{H}) | a = 1, \cdots, 2n\}, \tag{18}$$

which is generated by the set of coordinate generators obeying the commutation relation

$$[y^a, y^b]_\star = i\theta^{ab}. \tag{19}$$

The $\star$-algebra (19) is related to the Moyal–Heisenberg algebra (12) where $p_a = B_{ab}y^b$ and $(\theta)^{ab} = (B^{-1})^{ab} = l_s^2(\mathbf{1}_n \otimes i\sigma^2)$ is a $2n \times 2n$ constant symplectic matrix. Let us denote the NC $\star$-algebra $\mathcal{A}_\theta$ generated by (19) as $\mathbb{R}_\theta^{2n}$.

Given a Hermitian operator $\widehat{\phi}_a(y) \in \mathcal{A}_\theta$, we have a matrix representation in $\mathcal{H}$ as follows:

$$\widehat{\phi}_a(y) = \sum_{n,m=1}^{\infty} |n\rangle\langle n|\widehat{\phi}_a(y)|m\rangle\langle m| = \sum_{n,m=1}^{\infty} (\phi_a)_{nm}|n\rangle\langle m| \tag{20}$$

using the completeness of $\mathcal{H}$, i.e., $\sum_{n=1}^{\infty} |n\rangle\langle n| = \mathbf{1}_\mathcal{H}$. The unitary representation of the operator algebra $\mathcal{A}_\theta$ can thus be understood as a linear transformation acting on an $N$-dimensional Hilbert space $\mathcal{H}_N$:

$$\mathcal{A}_\theta : \mathcal{H}_N \to \mathcal{H}_N. \tag{21}$$

That is, we have the identification [56,57]

$$\mathcal{A}_N \cong \mathrm{End}(\mathcal{H}_N) \cong \mathcal{A}_\theta. \tag{22}$$

As a result, the inner automorphism $\mathrm{Inn}(\mathcal{A}_N)$ of the matrix algebra $\mathcal{A}_N$ is translated into that of the NC $\star$-algebra $\mathcal{A}_\theta$, denoted by $\mathrm{Inn}(\mathcal{A}_\theta)$. Its infinitesimal generators consist of an inner derivation $\mathfrak{D}$ defined by the map [25–28]

$$\mathcal{A}_\theta \to \mathfrak{D} : \mathcal{O} \mapsto \mathrm{ad}_\mathcal{O} = -i[\mathcal{O}, \cdot]_\star \tag{23}$$

for any operator $\mathcal{O} \in \mathcal{A}_\theta$. Using the Jacobi identity of the NC $\star$-algebra $\mathcal{A}_\theta$, one can easily verify the Lie algebra homomorphism:

$$[\mathrm{ad}_{\mathcal{O}_1}, \mathrm{ad}_{\mathcal{O}_2}] = -i\mathrm{ad}_{[\mathcal{O}_1, \mathcal{O}_2]_\star} \tag{24}$$

for any $\mathcal{O}_1, \mathcal{O}_2 \in \mathcal{A}_\theta$. In particular, we are interested in the set of derivations determined by NC gauge fields in Equation (18):

$$\{\widehat{V}_a \equiv \mathrm{ad}_{\widehat{\phi}_a} \in \mathfrak{D} | \widehat{\phi}_a(y) = p_a + \widehat{A}_a(y) \in \mathcal{A}_\theta, \ a = 1, \cdots, 2n\}. \tag{25}$$

In a large-distance limit, i.e., $|\theta| \to 0$, one can expand the NC vector fields $\widehat{V}_a$ using the explicit form of the Moyal $\star$-product. The result takes the form

$$\widehat{V}_a = V_a^\mu(y)\frac{\partial}{\partial y^\mu} + \sum_{p=2}^{\infty} V_a^{\mu_1\cdots\mu_p}(y)\frac{\partial}{\partial y^{\mu_1}}\cdots\frac{\partial}{\partial y^{\mu_p}} \in \mathfrak{D}. \tag{26}$$

Thus, the NC vector fields in $\mathfrak{D}$ generate an infinite tower of the so-called polyvector fields [27]. Note that the leading term gives rise to the ordinary vector fields that will be identified with a frame basis associated to the tangent bundle $T\mathcal{M}$ of an emergent manifold $\mathcal{M}$. If the leading term in Equation (26) already generated the gravitational fields of spin 2, the higher-order terms would correspond to higher-spin fields with spin $\geq 3$.

Since we have started with a large $N$ matrix model, it is natural to expect that the IKKT-type matrix model (6) is dual to a higher-dimensional gravity or string theory according to the large $N$ duality or gauge/gravity duality [58]. The emergent gravity is realized via the gauge/gravity duality as follows [27]:

$$\mathcal{A}_N \implies \mathcal{A}_\theta \implies \mathfrak{D}. \tag{27}$$

The gauge theory side of the duality is described by the set of large $N$ matrices that consists of an associative, but NC, algebra $\mathcal{A}_N$. By choosing a proper vacuum such as Equation (9), a matrix in $\mathcal{A}_N$ is regarded as a linear representation of an operator acting on a separable Hilbert space $\mathcal{H}$. That is, the matrix algebra $\mathcal{A}_N$ is realized as a linear representation of an operator algebra $\mathcal{A}_\theta$ on the Hilbert space $\mathcal{H}$, i.e., $\mathcal{A}_N \cong \mathrm{End}(\mathcal{H})$. Consequently, the algebra $\mathcal{A}_N$ is isomorphically mapped to the NC $\star$-algebra $\mathcal{A}_\theta$, as Equation (20) has clearly illustrated. The gravity side of the duality is defined by associating the derivation $\mathfrak{D}$ of the

algebra $\mathcal{A}_\theta$ with a quantized frame bundle $\widehat{\mathfrak{X}}(\mathcal{M})$ of an emergent spacetime manifold $\mathcal{M}$. The noncommutativity of an underlying algebra is, thus, crucial to realize the emergent gravity. This is the reason why we need the Moyal–Heisenberg vacuum (10) instead of the conventional Coulomb branch vacuum [39]. After all, in order to describe a quantum geometry properly, it is necessary to distinguish two types of vacuum in the Coulomb branch: diagonalizable vs. nondiagonalizable vacua.

At this stage it is important to understand how (local) coordinates which have been used to define the vector fields in $\mathfrak{D}$ arise from matrices in $\mathcal{A}_N$. The crux is the isomorphism (22) between the matrix algebra $\mathcal{A}_N$ and the NC $\star$-algebra $\mathcal{A}_\theta$ in the limit $N \to \infty$. Here, the quantity $|\theta|$ in (19) plays a role similar to $\hbar$ in quantum mechanics. Therefore, we will obtain a classical algebra $C^\infty(\mathcal{M})$ generated by smooth functions on $\mathcal{M}$ from the NC $\star$-algebra $\mathcal{A}_\theta$ when we take a commutative limit, $|\theta| \to 0$. Then, given an open set $U \subset \mathcal{M}$, one can use some local functions $(y^1, \cdots, y^{2n}) : U \to \mathbb{R}^{2n}$ to define a coordinate chart around $p \in U$. Since the underlying functions are smooth, one can introduce infinitesimal quantities such as tangent vectors $\frac{\partial}{\partial y^\mu}|_p$ and covectors $dy^\mu|_p$ at $p \in U$ associated with the given coordinate system. Note that, if we had chosen a diagonalized vacuum (see footnote 6) instead of the nondiagonalizable vacuum (10), the existence of such continuous variables and infinitesimal values would not be guaranteed even in the limit $N \to \infty$.

Recognizing the intrinsic locality is crucial when grasping the emergence of geometry through the duality chain in Equation (27). It is necessary to consider patching or gluing together the local constructions to form a set of global quantities. For this purpose, the concept of sheaf may be essential because it makes it possible to reconstruct global data starting from open sets of locally defined data [59]. We provide a succinct overview of this feature, as it has already been comprehensively discussed in Ref. [27]. Its characteristic feature becomes transparent when the commutative limit, i.e., $|\theta| \to 0$, is taken into account. In this limit, the NC $\star$-algebra $\mathcal{A}_\theta$ reduces to a Poisson algebra $\mathfrak{P}^{(i)} = (C^\infty(U_i), \{-, -\}_\theta)$ defined on a local patch $U_i \subset M$ in an open covering $M = \bigcup_{i \in I} U_i$.[7] The Poisson algebra $\mathfrak{P}^{(i)}$ arises as follows. Let $L \to M$ be a line bundle over $M$ whose connection is denoted by $\mathcal{A}$. We assume that the curvature $\mathcal{F}$ of the line bundle $L$ is a *nondegenerate*, closed two-form. Therefore, we identify the curvature two-form $\mathcal{F} = d\mathcal{A}$ with a symplectic structure of $M$. On an open neighborhood $U_i \subset M$, it is possible to represent $\mathcal{F}^{(i)} = B + F^{(i)}$ where $F^{(i)} = dA^{(i)}$ and $B$ is the constant symplectic two-form already introduced in Equation (10). Consider a chart $(U_i, \phi_{(i)})$ where $\phi_{(i)} \in \mathrm{Diff}(U_i)$ is a local trivialization of the line bundle $L$ over the open subset $U_i$ obeying $\phi^*_{(i)}(\mathcal{F}^{(i)}) = B$. A local chart is guaranteed to exist thanks to either the Darboux theorem or the Moser lemma in symplectic geometry [60,61] and the local coordinate chart obeying $\phi^*_{(i)}(\mathcal{F}^{(i)}) = B$ is called Darboux coordinates. Thus, the line bundle $L \to M$ corresponds to a dynamical symplectic manifold $(M, \mathcal{F})$ where $\mathcal{F} = B + dA$. The dynamical system is locally described by the Poisson algebra $\mathfrak{P}^{(i)} = (C^\infty(U_i), \{-, -\}_\theta)$ in which the vector space $C^\infty(U_i)$ is formed by the set of Darboux transformations $\phi_{(i)} \in \mathrm{Diff}(U_i)$ equipped with the Poisson bracket defined by the Poisson bivector $\theta = B^{-1} \in \Gamma(\Lambda^2 TM)$.

Consider a collection of local charts to make an atlas $\{(U_i, \phi_{(i)})\}$ on $M = \bigcup_{i \in I} U_i$ and complete the atlas by gluing these charts on their overlaps. To be precise, suppose that $(U_i, \phi_{(i)})$ and $(U_j, \phi_{(j)})$ are two coordinate charts and $F^{(i)} = dA^{(i)}$ and $F^{(j)} = dA^{(j)}$ are local curvature two-forms on $U_i$ and $U_j$, respectively. We choose the coordinate maps $\phi_{(i)} \in \mathrm{Diff}(U_i)$ and $\phi_{(j)} \in \mathrm{Diff}(U_j)$ such that $\phi^*_{(i)}(B + F^{(i)}) = B$ and $\phi^*_{(j)}(B + F^{(j)}) = B$. On an intersection $U_i \cap U_j$, the local data $(A^{(i)}, \phi_{(i)})$ and $(A^{(j)}, \phi_{(j)})$ on Darboux charts $(U_i, \phi_{(i)})$ and $(U_j, \phi_{(j)})$, respectively, are glued together by [62,63]

$$A^{(j)} = A^{(i)} + d\lambda^{(ji)}, \tag{28}$$

$$\phi_{(ji)} = \phi_{(j)} \circ \phi_{(i)}^{-1}, \tag{29}$$

where $\phi_{(ji)} \in \text{Diff}(U_i \cap U_j)$ is a symplectomorphism on $U_i \cap U_j$ generated by a Hamiltonian vector field $X_{\lambda^{(ji)}}$ satisfying $\iota(X_{\lambda^{(ji)}})B + d\lambda^{(ji)} = 0$. We sometimes denote the interior product $\iota_X$ by $\iota(X)$ for a notational convenience. Similarly, we can glue the local Poisson algebras $\mathfrak{P}^{(i)}$ to form a globally defined Poisson algebra $\mathfrak{P} = \bigcup_{i \in I} \mathfrak{P}^{(i)}$. The global vector fields $V_a = V_a^\mu(y)\frac{\partial}{\partial y^\mu} \in \Gamma(T\mathcal{M})$, $a = 1, \cdots, 2n$, in Equation (26) can be obtained by applying a similar globalization to the derivation $\mathfrak{D}$, which form a linearly independent basis of the tangent bundle $T\mathcal{M}$ of a $2n$-dimensional emergent manifold $\mathcal{M}$. As a consequence, the set of global vector fields $\mathfrak{X}(\mathcal{M}) = \{V_a | a = 1, \cdots, 2n\}$ results from the globally defined Poisson algebra $\mathfrak{P}$ [27].

The vector fields $V_a \in \mathfrak{X}(\mathcal{M})$ are related to an orthonormal frame, the so-called vielbeins $E_a \in \Gamma(T\mathcal{M})$, in general relativity by the relation

$$V_a = \lambda E_a, \qquad a = 1, \cdots, 2n. \tag{30}$$

The conformal factor $\lambda \in C^\infty(\mathcal{M})$ is determined by imposing the condition that the vector fields $V_a$ preserve a volume form

$$\nu = \lambda^2 v^1 \wedge \cdots \wedge v^{2n}, \tag{31}$$

where $v^a = v_\mu^a(y)dy^\mu \in \Gamma(T^*\mathcal{M})$ are coframes dual to $V_a$, i.e., $\langle v^a, V_b \rangle = \delta_b^a$. This means that the vector fields $V_a$ obey the conditions

$$\mathcal{L}_{V_a}\nu = \left(\nabla \cdot V_a + (2 - 2n)V_a \ln \lambda\right)\nu = 0, \qquad \forall a = 1, \cdots, 2n, \tag{32}$$

where $\mathcal{L}_X = \iota_X d + d\iota_X$ is the Lie derivative with respect to a vector field $X$. Note that a symplectic manifold always admits such volume-preserving vector fields (see Appendix B in [27]). Together with the volume-preserving condition (32), the relation (30) completely determines a $2n$-dimensional Riemannian manifold $\mathcal{M}$ whose metric is given by [25–27]

$$\begin{aligned} ds^2 &= \mathcal{G}_{\mu\nu}(x)dx^\mu \otimes dx^\nu = e^a \otimes e^a \\ &= \lambda^2 v^a \otimes v^a = \lambda^2 v_\mu^a(y)v_\nu^a(y)dy^\mu \otimes dy^\nu, \end{aligned} \tag{33}$$

where $e^a = e_\mu^a(x)dx^\mu = \lambda v^a \in \Gamma(T^*\mathcal{M})$ are orthonormal one-forms on $\mathcal{M}$. After all, the $2n$-dimensional Riemannian manifold $\mathcal{M}$ is emergent from the commutative limit of polyvector fields $\widehat{V}_a = V_a + \mathcal{O}(\theta^2) \in \mathfrak{D}$ derived from NC $U(1)$ gauge fields.

So far, we have discussed the emergence of spaces only. However, the theory of relativity dictates that space and time must be coalesced into the form of Minkowski spacetime in a locally inertial frame. Hence, if general relativity is realized from an NC $\star$-algebra $\mathcal{A}_\theta$, it is necessary to put space and time on an equal footing in the NC $\star$-algebra $\mathcal{A}_\theta$. If space is emergent, so should time be. Thus, an important problem is how to realize the emergence of "time". However, any physical theory that we know does not treat time as a dynamical variable. Therefore, we assert that the concept of emergent time needs to be understood differently from emergent spaces (we will later discuss a perplexing problem that arises when we promote time to a "dynamical" variable). Quantum mechanics imparts a valuable insight, emphasizing the intricate relationship between the definition of (particle) time and the dynamics inherent in the system. In quantum mechanics, the time evolution of a dynamical system is defined as an inner automorphism of NC algebra $\mathcal{A}_\hbar$ generated by the NC phase space

$$[x^i, x^j] = 0, \qquad [x^i, p_j] = i\hbar\delta_j^i, \qquad i, j = 1, \cdots, n. \tag{34}$$

The time evolution for an observable $f \in \mathcal{A}_\hbar$ is simply an inner derivation of $\mathcal{A}_\hbar$ given by

$$\frac{df}{dt} = \frac{i}{\hbar}[H, f], \tag{35}$$

where $H$ is a Hamiltonian operator of the dynamical system and will be identified with a temporal gauge field $A_0$, i.e., $H = -A_0$, in matrix quantum mechanics. The integral of Equation (35) is simply a unitary transformation of the observable $f \in \mathcal{A}_\hbar$:

$$f(t) = U(t)f(0)U(t)^\dagger, \tag{36}$$

where $U(t) = e^{\frac{iHt}{\hbar}}$ is a unitary operator. For a quantum dynamical system that has a classical analogue, Equation (36) implies that unitary transformations in the quantum theory are an analogue of canonical (or contact) transformations in the classical theory (see Section 26 Unitary transformations in [55]).

Given a symplectic form $\omega = \sum_{i=1}^{n} dx^i \wedge dp_i$ on phase space, one can introduce a Hamiltonian vector field $X_H$ defined by $\iota_{X_H} \omega = dH$. The one-parameter family of canonical transformations can then be thought of as "Hamiltonian flow" on phase space:

$$\left( X^i(x, p; t) = x^i + tX_H(x^i), P_i(x, p; t) = p_i + tX_H(p_i) \right). \tag{37}$$

According to this active viewpoint, the canonical transformation takes one point in the phase space, $(x^i, p_i)$, to another point in the same phase space, $\left( X^i(x, p; t), P_i(x, p; t) \right)$. Correspondingly, the point at time $t$ can be understood as a one-parameter family of deformations (or changes) generated by a smooth function $H = H(x, p)$. We will define the concept of emergent time based on this perspective.

A remarkable picture, as observed by Feynman [64], Souriau and Sternberg [65], is that the physical forces such as the electromagnetic, weak and strong forces, can be realized as the deformations of an underlying vacuum algebra such as Equation (34). For example, the most general deformation of the Heisenberg algebra (34) within the *associative* algebra $\mathcal{A}_\hbar$ is given by

$$x^i \to x^i, \qquad p_i \to p_i + A_i(x, t), \qquad H \to H + A_0(x, t), \tag{38}$$

where $(A_0, A_i)(x, t)$ must be electromagnetic gauge fields. Then, the time evolution of a particle system under a time-dependent external force is given by

$$\frac{df}{dt} = \frac{\partial f}{\partial t} + \frac{i}{\hbar}[H, f]. \tag{39}$$

Note that the construction of the NC algebra $\mathcal{A}_N$ or $\mathcal{A}_\theta$ bears a close parallel to quantum mechanics. The former is based on the NC space (19), while the latter is based on the NC phase space (34). The NC $U(1)$ gauge fields in Equation (11) act as deformations of the vacuum algebra (10) in the matrix algebra $\mathcal{A}_N$, similarly to Equation (38) in the quantum algebra $\mathcal{A}_\hbar$. Therefore, we can apply the same philosophy to the NC algebra $\mathcal{A}_N$ or $\mathcal{A}_\theta$ to define a dynamical system based on the Moyal–Heisenberg algebra (10). In other words, we can consider a one-parameter family of deformations of zero-dimensional matrices which is parameterized by the coordinate $t$. Then, the one-parameter family of deformations characterized by (11) can be regarded as the time evolution of a dynamical system. For this purpose, we extend the NC algebra $\mathcal{A}_\theta$ to $\mathcal{A}_\theta^1 \equiv \mathcal{A}_\theta\big(C^\infty(\mathbb{R})\big) = C^\infty(\mathbb{R}) \otimes \mathcal{A}_\theta$ whose generic element takes the form

$$\widehat{f}(t, y) \in \mathcal{A}_\theta^1. \tag{40}$$

The matrix representation (20) is then replaced by

$$\widehat{f}(t, y) = \sum_{n,m=1}^{\infty} |n\rangle\langle n|\widehat{f}(t, y)|m\rangle\langle m| = \sum_{n,m=1}^{\infty} f_{nm}(t)|n\rangle\langle m| \tag{41}$$

where $f_{nm}(t) := [f(t)]_{nm}$ are elements of a matrix $f(t)$ in $\mathcal{A}_N^1 \equiv \mathcal{A}_N\big(C^\infty(\mathbb{R})\big) = C^\infty(\mathbb{R}) \otimes \mathcal{A}_N$ as a representation of Equation (40) on the Hilbert space (17). As the Heisenberg

Equation (39) in quantum mechanics suggests, the evolution equation for an observable $\widehat{f}(t,y) \in \mathcal{A}_\theta^1$ in the Heisenberg picture is defined by

$$\frac{d\widehat{f}(t,y)}{dt} = \frac{\partial \widehat{f}(t,y)}{\partial t} - i[\widehat{A}_0(t,y), \widehat{f}(t,y)]_\star \equiv \widehat{D}_0 \widehat{f}(t,y) \tag{42}$$

where we denoted the local Hamiltonian density by

$$\widehat{H}(t,y) \equiv -\widehat{A}_0(t,y) \in \mathcal{A}_\theta^1. \tag{43}$$

Note that

$$-i[\phi_a, \widehat{f}(t)] = \partial_a \widehat{f}(t,y) - i[\widehat{A}_a(t,y), \widehat{f}(t,y)]_\star \equiv \widehat{D}_a \widehat{f}(t,y), \tag{44}$$

where the representation (41) has been employed. Then, one can see that the inner automorphism $\mathrm{Inn}(\mathcal{A}_\theta)$ of $\mathcal{A}_\theta$ can be lifted to the automorphism of $\mathcal{A}_\theta^1$ given by

$$\widehat{A}_0(t,y) \to \widehat{U}(t,y) \star \frac{\partial \widehat{U}^{-1}(t,y)}{\partial t} + \widehat{U}(t,y) \star \widehat{A}_0(t,y) \star \widehat{U}^{-1}(t,y), \tag{45}$$

$$\widehat{A}_a(t,y) \to \widehat{U}(t,y) \star \frac{\partial \widehat{U}^{-1}(t,y)}{\partial y^a} + \widehat{U}(t,y) \star \widehat{A}_a(t,y) \star \widehat{U}^{-1}(t,y), \tag{46}$$

where $\widehat{U}(t,y) = e_\star^{i\widehat{\lambda}(t,y)}$ with $\widehat{\lambda}(t,y) \in \mathcal{A}_\theta^1$. It is obvious that the above automorphism is nothing but the gauge transformation for NC $U(1)$ gauge fields in $(2n+1)$-dimensions [66].

Our leitmotif is that a consistent theory of quantum gravity should be background-independent, so that it should not presuppose any spacetime background on which fundamental processes develop. Hence, the background-independent theory must provide a mechanism of spacetime generation such that every spacetime structure including the flat spacetime arises as a solution of the theory itself. A zero-dimensional matrix model such as Equation (6) is the most natural candidate for such a background-independent theory because it does not have to assume the prior existence of spacetime to define the theory.

Then, how can Minkowski spacetime also emerge as a solution of an underlying background-independent theory? We emphasize again that the NC nature of the vacuum solution, e.g., Equation (10), is essential to realize the large $N$ duality via the duality chain (27). A profound feature is that the background-independent theory is intrinsically dynamical because the space of all possible solutions is generated by generic deformations of a primitive vacuum such as Equation (10) [27]. We contend that the dynamics governed by the Moyal–Heisenberg vacuum (9) is characterized by the NC algebra $\mathcal{A}_N^1 = \mathcal{A}_N(C^\infty(\mathbb{R})) = C^\infty(\mathbb{R}) \otimes \mathcal{A}_N$. One may regard $\mathcal{A}_N^1$ as a one-parameter family of deformations of the algebra $\mathcal{A}_N$. In this case, we can generalize the duality chain (27) to realize the "time-dependent" gauge/gravity duality as follows:

$$\mathcal{A}_N^1 \implies \mathcal{A}_\theta^1 \implies \mathfrak{D}^1. \tag{47}$$

It is well known [67] that in the case of $\mathcal{A}_N^1$ or $\mathcal{A}_\theta^1$, the module of its derivations can be written as a direct sum of the submodules of horizontal and inner derivations:

$$\mathfrak{D}^1 = \mathrm{Hor}(\mathcal{A}_N^1) \oplus \mathfrak{D}(\mathcal{A}_N^1) \cong \mathrm{Hor}(\mathcal{A}_\theta^1) \oplus \mathfrak{D}(\mathcal{A}_\theta^1) \tag{48}$$

where horizontal derivation is a lifting of smooth vector fields on $\mathbb{R}$ onto $\mathcal{A}_N^1$ or $\mathcal{A}_\theta^1$ and is locally generated by a vector field

$$g(t,y) \frac{\partial}{\partial t} \in \mathrm{Hor}(\mathcal{A}_\theta^1). \tag{49}$$

The inner derivation $\mathfrak{D}(\mathcal{A}^1_\theta)$ is defined by lifting the NC vector fields in Equation (25) onto $\mathcal{A}^1_\theta$ and generated by

$$\left\{\widehat{V}_a(t) \equiv \mathrm{ad}_{\widehat{\phi}_a} \in \mathfrak{D}(\mathcal{A}^1_\theta) | \widehat{\phi}_a(t,y) = p_a + \widehat{A}_a(t,y) \in \mathcal{A}^1_\theta, \ a = 1, \cdots, 2n\right\} \tag{50}$$

and

$$\left\{\widehat{V}_0(t) - \frac{\partial}{\partial t} \equiv \mathrm{ad}_{\widehat{A}_0} \in \mathfrak{D}(\mathcal{A}^1_\theta) | \widehat{A}_0(t,y) \in \mathcal{A}^1_\theta\right\}. \tag{51}$$

It might be remarked that the definition of the time-like vector field $\widehat{V}_0(t)$ is motivated by the quantum Hamilton's Equation (42), i.e.,

$$\widehat{V}_0(t) := \frac{d}{dt}. \tag{52}$$

Consequently, the module of the derivations of the NC algebra $\mathcal{A}^1_\theta$ is given by

$$\mathfrak{D}^1 = \left\{\widehat{V}_A(t) = (\widehat{V}_0, \widehat{V}_a)(t) | \widehat{V}_0(t) = \frac{\partial}{\partial t} + \mathrm{ad}_{\widehat{A}_0}, \ \widehat{V}_a(t) = \mathrm{ad}_{\widehat{\phi}_a}, \ A = 0, 1, \cdots, 2n\right\}. \tag{53}$$

In the commutative limit, $|\theta| \to 0$, the time-dependent polyvector fields $\widehat{V}_A(t)$ in $\mathfrak{D}^1$ take the following form

$$\widehat{V}_0(t) = \frac{\partial}{\partial t} + A^\mu_0(t,y)\frac{\partial}{\partial y^\mu} + \sum_{p=2}^\infty A^{\mu_1 \cdots \mu_p}_0(t,y)\frac{\partial}{\partial y^{\mu_1}} \cdots \frac{\partial}{\partial y^{\mu_p}}, \tag{54}$$

$$\widehat{V}_a(t) = V^\mu_a(t,y)\frac{\partial}{\partial y^\mu} + \sum_{p=2}^\infty V^{\mu_1 \cdots \mu_p}_a(t,y)\frac{\partial}{\partial y^{\mu_1}} \cdots \frac{\partial}{\partial y^{\mu_p}}. \tag{55}$$

Let us truncate the above polyvector fields to ordinary vector fields given by

$$\mathfrak{X}(\mathcal{M}) = \left\{V_A = V^M_A(t,y)\frac{\partial}{\partial X^M} | A, M = 0, 1, \cdots, 2n\right\} \tag{56}$$

where $V^0_A = \delta^0_A$ and $X^M = (t, y^\mu)$ are local coordinates on an emergent *Lorentzian* manifold $\mathcal{M}$ of $(2n+1)$-dimensions. The orthonormal vielbeins on $T\mathcal{M}$ are then obtained by the prescription

$$(V_0, V_a) = (E_0, \lambda E_a) \in \Gamma(T\mathcal{M}). \tag{57}$$

The dual orthonormal basis on $T^*\mathcal{M}$ is defined by the relation $\langle v^A, V_B \rangle = \delta^A_B$ and it is given by $v^A = (v^0, v^a) = \left(dt, v^a_\mu(dy^\mu - A^\mu_0(t,y))\right)$ where $v^a_\mu V^\mu_b = \delta^a_b$. From Equation (57), we have

$$(e^0, e^a) = (v^0, \lambda v^a) \in \Gamma(T^*\mathcal{M}). \tag{58}$$

The conformal factor $\lambda \in C^\infty(\mathcal{M})$ is similarly determined by the volume-preserving condition

$$\mathcal{L}_{V_A} v_t = \left(\nabla \cdot V_A + (2 - 2n)V_A \ln \lambda\right)v_t = 0, \qquad \forall A = 0, 1, \cdots, 2n. \tag{59}$$

The above condition explicitly reads as

$$\frac{\partial \rho}{\partial t} + \partial_\mu(\rho A^\mu_0) = 0 \quad \& \quad \partial_\mu(\rho V^\mu_a) = 0, \tag{60}$$

where $\rho = \lambda^2 \det v^a_\mu$ and

$$v_t \equiv dt \wedge v = \lambda^2 dt \wedge v^1 \wedge \cdots \wedge v^{2n} \tag{61}$$

is a $(2n+1)$-dimensional volume form on $\mathcal{M}$. If the structure equation of vector fields $V_A \in \Gamma(T\mathcal{M})$ is defined by

$$[V_A, V_B] = -g_{AB}{}^C V_C, \tag{62}$$

the volume-preserving condition (59) can equivalently be written as [26,39]

$$g_{BA}{}^B = V_A \ln \lambda^2. \tag{63}$$

In the end, the Lorentzian metric on a $(2n+1)$-dimensional spacetime manifold $\mathcal{M}$ is given by [25,27]

$$
\begin{aligned}
ds^2 &= \mathcal{G}_{MN}(X) dX^M \otimes dX^N = \eta_{AB} e^A \otimes e^B \\
&= -v^0 \otimes v^0 + \lambda^2 v^a \otimes v^a = -dt^2 + \lambda^2 v^a_\mu v^a_\nu (dy^\mu - \mathbf{A}^\mu)(dy^\nu - \mathbf{A}^\nu)
\end{aligned} \tag{64}
$$

where $\mathbf{A}^\mu := A_0^\mu(t,y)dt$.

It should be noted that the time evolution (52) for a general time-dependent system is not completely generated by an inner automorphism since $\mathrm{Hor}(\mathcal{A}_\theta^1)$ is not an inner derivation but instead an outer derivation. This happens since the time variable $t$ is single. Thus, one may extend the phase space by introducing a conjugate variable $H$ of $t$ so that the extended phase space becomes a symplectic manifold. Then, it is well known [60,61] that the time evolution of a time-dependent system can be defined by the inner automorphism of the extended phase space whose extended Poisson bivector is given by

$$\vartheta = \theta + \frac{\partial}{\partial t} \wedge \frac{\partial}{\partial H} \tag{65}$$

where

$$\theta = \frac{1}{2}\theta^{\mu\nu} \frac{\partial}{\partial y^\mu} \wedge \frac{\partial}{\partial y^\nu} \tag{66}$$

is the original Poisson bivector related to the NC space (19). As a result, one can see [26] that the temporal vector field (52) is realized as a generalized Hamiltonian vector field defined by

$$V_0 = \mathcal{X}_H = -\vartheta(dH - dA_0) = \frac{\partial}{\partial t} + X_H \tag{67}$$

where $X_H = \theta(dA_0)$ is the original Hamiltonian vector field which is a classical part of the inner derivation $\mathrm{ad}_{\widehat{A}_0} = X_H + \mathcal{O}(\theta^2) \in \mathfrak{D}(\mathcal{A}_\theta^1)$. However, we must bear the cost associated with the extension of the phase space. In the extended phase space, the time $t$ is now promoted to a dynamical variable, whereas it was simply an affine parameter describing a Hamiltonian flow in the old phase space. Then, the extended Poisson structure (65) raises a serious issue as to whether the time variable for a general time-dependent system might also be quantized; in other words, time also becomes an operator obeying the commutation relation $[t, H] = -i$. Then, it becomes difficult to defend the causality of physical theories. We want to refrain from addressing this abstruse issue since it persists as a challenging open problem, even within the realm of quantum mechanics.

We address the time issue through a more pragmatic approach.[8] In mechanical systems, time is defined through a contact structure [68,69]. Suppose that $(M, B \equiv \theta^{-1})$ is the original symplectic manifold. Now, we consider a contact manifold $(\mathbb{R} \times M, \widetilde{B})$ where $\widetilde{B} = \pi_2^* B$ is defined by the projection $\pi_2 : \mathbb{R} \times M \to M$, $\pi_2(t,x) = x$ [61]. We define the concept of spacetime in emergent gravity through the contact manifold $(\mathbb{R} \times M, \widetilde{B})$ in the sense that the derivations in Equation (53) can be obtained by quantizing the contact manifold $(\mathbb{R} \times M, \widetilde{B})$. Indeed, it is shown in Appendix A that the time-like vector field $V_0$ in Equation (67) arises as a Hamiltonian vector field of a cosymplectic manifold whose particular class is a contact manifold. Note that the emergent geometry described by the metric (64) respects the (local) Lorentz symmetry. One can see that the Lorentzian manifold $\mathcal{M}$ becomes the Minkowski spacetime on a local Darboux chart in which all fluctuations die out, i.e., $v^a_\mu \to \delta^a_\mu$, $\mathbf{A}^\mu \to 0$, so $\lambda \to 1$. We have to emphasize that the vacuum algebra responsible for the emergence of

the Minkowski spacetime is the Moyal–Heisenberg algebra (10). Many surprising results will immediately come from this dynamical origin of the flat spacetime [23,25,26], which is absent in general relativity.

We close this section by observing that the quantized version of the contact manifold $(\mathbb{R} \times M, \widetilde{B})$ is described by a matrix quantum mechanics (MQM) whose action is given by

$$S = \frac{1}{g_{YM}^2} \int dt \mathrm{Tr} \left( \frac{1}{2} (D_0 \phi_a)^2 + \frac{1}{4} [\phi_a, \phi_b]^2 \right), \tag{68}$$

where $D_0 \phi_a = \frac{\partial \phi_a}{\partial t} - i[A_0, \phi_a]$. The equations of motion for the matrix action (68) are given by

$$D_0^2 \phi_a + [\phi_b, [\phi_a, \phi_b]] = 0, \tag{69}$$

which must be supplemented with the Gauss constraint

$$[\phi_a, D_0 \phi_a] = 0. \tag{70}$$

We interpret the matrix model (68) as a Hamiltonian system of the IKKT matrix model whose action is given by Equation (6). Note that the original BFSS matrix model [70] contains nine adjoint scalar fields, while Equation (68) has an even number of adjoint scalar fields. For the former case, we do not know how to realize the adjoint scalar fields as a matrix representation of NC $U(1)$ gauge fields on a Hilbert space such as (41). It may even be nontrivial to construct the Hilbert space because the M-theory is involved with a 3-form instead of symplectic 2-form. For the latter case, on the other hand, the previous Moyal–Heisenberg vacuum (9) is naturally extended to the vacuum configuration of $\mathcal{A}_N^1$ given by

$$\langle \phi_a \rangle_{\mathrm{vac}} = p_a, \qquad \langle A_0 \rangle_{\mathrm{vac}} = -\mathcal{E}, \tag{71}$$

where the vacuum moduli $p_a \in \mathcal{A}_N^1$ satisfy the commutation relation (10) and $\mathcal{E}$ is a constant vacuum energy density proportional to the identity matrix. We consider all possible deformations of the vacuum (71) and parameterize them as

$$\widehat{\phi}_A(t, y) = p_A + \widehat{A}_A(t, y) \in \mathcal{A}_\theta^1, \tag{72}$$

where $p_0 = i \frac{\partial}{\partial t} - \mathcal{E}$ and the isomorphism (41) between $\mathcal{A}_N^1$ and $\mathcal{A}_\theta^1$ was used. Note that

$$[\widehat{\phi}_A, \widehat{\phi}_B]_\star = -i(B_{AB} - \widehat{F}_{AB}), \tag{73}$$

where

$$\widehat{F}_{AB} = \partial_A \widehat{A}_B - \partial_B \widehat{A}_A - i[\widehat{A}_A, \widehat{A}_B]_\star \in \mathcal{A}_\theta^1 \tag{74}$$

and

$$B_{AB} = \begin{pmatrix} 0 & 0 \\ 0 & B_{ab} \end{pmatrix}.$$

Plugging the fluctuations (72) into Equation (68) leads to a $(2n + 1)$-dimensional NC $U(1)$ gauge theory with the action [54,57]

$$\begin{aligned} S &= \frac{1}{g_{YM}^2} \int dt \mathrm{Tr} \left( \frac{1}{2} (D_0 \phi_a)^2 + \frac{1}{4} [\phi_a, \phi_b]^2 \right) \\ &= -\frac{1}{4 G_{YM}^2} \int d^{2n+1} y \left( \widehat{F}_{AB} - B_{AB} \right)^2, \end{aligned} \tag{75}$$

where $G_{YM}^2 = (2\pi)^n |\mathrm{Pf}\theta| g_{YM}^2$ is the $(2n+1)$-dimensional gauge coupling constant. By applying the duality chain (47) to time-dependent matrices in $\mathcal{A}_N^1$, it is straightforward to derive the module $\mathfrak{D}^1$ in Equation (53) from the large $N$ matrices or NC $U(1)$ gauge fields in the action (75). A Lorentzian spacetime described by the metric (64) corresponds to a classical geometry derived from the NC module $\mathfrak{D}^1$ [27].

### 3. Cosmic Inflation from Time-Dependent Matrices

From now on, we will focus on the matrix quantum mechanics (MQM) to address the background-independent formulation of cosmic inflation. Let us rewrite the action (68) in the form

$$S = \frac{1}{4g^2} \int dt \, \eta^{AC} \eta^{BD} \text{Tr}[\phi_A, \phi_B][\phi_C, \phi_D], \tag{76}$$

where $\phi_0 \equiv iD_0 = i\frac{\partial}{\partial t} + A_0(t)$, $\phi_A(t) = (\phi_0, \phi_a)(t)$, and $\eta^{AB} = \text{diag}(-1, 1, \cdots, 1)$, $A, B = 0, 1, \cdots, 2n$. With the definition of the symbol $\eta^{AB}$, it is easy to see that the matrix action (76) has a global automorphism given by

$$\phi_A \to \phi'_A = \Lambda_A{}^B \phi_B + c_A \tag{77}$$

if $\Lambda_A{}^B$ is a rotation in $SO(2n, 1)$ and $c_A$ are constants proportional to the identity matrix. It will be shown later that the global symmetry (77) is responsible for the Poincaré symmetry of flat spacetime emergent from a vacuum in the Coulomb branch of MQM and so will be called the Poincaré automorphism. We remark that the time $t$ in the action (76) is not a dynamical variable but an affine parameter. The concept of emergent time was defined in the previous section by considering a one-parameter family of deformations of zero-dimensional matrices which is parameterized by the coordinate $t$. The one-parameter family of deformations can then be regarded as the time evolution of a dynamical system. In this context, the one-dimensional matrix model (76) can be interpreted as a Hamiltonian system of a zero-dimensional (e.g., IKKT) matrix model [27]. A close analogy with quantum mechanics implies that the concept of emergent time is derived from the time evolution of the dynamical system. Although spatial coordinates and time are introduced in different ways, Equation (77) implies that they are connected by Lorentz transformations and coalesced into the form of Minkowski spacetime in a locally inertial frame.

The duality chain (47) implies that the gravitational variables such as vielbeins in general relativity arise from the commutative limit of NC $U(1)$ gauge fields. Then, one may ask where the Minkowski spacetime comes from. Let us look at the metric (64) to identify the origin of the Minkowski spacetime. Definitely, the Lorentzian manifold $\mathcal{M}$ becomes the Minkowski spacetime when all fluctuations die out, i.e., $v_b^a \to \delta_b^a$, $\mathbf{A}^a \to 0$ (and so $\lambda \to 1$). Therefore, the vacuum geometry for the metric (64) was originated from the vacuum configuration (71). In other words, the $(2n + 1)$-dimensional Minkowski spacetime emerges from the vacuum condensate (71) since the corresponding vielbeins and the metric are given by $E_A^{(0)} = V_A^{(0)} = \left(\frac{\partial}{\partial t}, \frac{\partial}{\partial y^a}\right)$ and $ds^2 = -dt^2 + d\mathbf{y} \cdot d\mathbf{y}$ [25,26]. The Minkowski spacetime originates from a coherent vacuum satisfying the Moyal–Heisenberg algebra (10), and the condensate (9) in the NC Coulomb vacuum induces a nontrivial vacuum energy density. We can calculate it using the action (75):

$$\rho_{\text{vac}} = \frac{1}{4G_{YM}^2} |B_{ab}|^2. \tag{78}$$

A striking fact is that the vacuum responsible for the generation of flat spacetime is not empty. Rather, the flat spacetime had originated from the uniform vacuum energy (78) known as the cosmological constant in general relativity. This is a tangible difference from Einstein gravity since Equation (1) enforces $T_{\mu\nu} = 0$ for the flat spacetime. Consequently, the emergent gravity reveals a remarkable picture in that a uniform vacuum energy such as Equation (78) does not gravitate (i.e., does not couple to gravity). As a result, the emergent gravity presents a striking contrast to general relativity. This important conclusion may be strengthened by applying the Lie algebra homomorphism (24) to the commutators in Equation (73), which reads as

$$-i\text{ad}_{[\phi_a, \phi_b]} \equiv \widehat{V}_{\widehat{F}_{ab} - B_{ab}} = \widehat{V}_{\widehat{F}_{ab}} = [\widehat{V}_a, \widehat{V}_b] \in \mathfrak{D}^1 \tag{79}$$

for a constant field strength $B_{ab}$. To stress clearly, the gravitational fields emergent from NC $U(1)$ gauge fields must be insensitive to the constant vacuum energy such as Equation (78). In the end, the emergent gravity clearly dismisses the notorious cosmological constant problem [23,25,26].

We observed that the MQM admits a global automorphism given by Equation (77). Let us see what the consequence of the Poincaré automorphism (77) is for the emergent spacetime geometry. The Poincaré automorphism leads to the transformation $V_A^{(0)} \rightarrow V_A^{'(0)} = \Lambda_A{}^B V_B^{(0)}$. However, this transformation does not change $\lambda^2$ because $\det \Lambda = 1$. The geometry for the transformed vacuum $p'_A$ is determined by the metric (64) that is still the Minkowski spacetime $\mathbb{R}^{2n,1}$. Therefore, we see that the vacuum responsible for the generation of flat spacetime is not unique but degenerates up to the Poincaré automorphism.[9] After all this, the global Poincaré symmetry of the Minkowski spacetime emerges from the Poincaré automorphism (77) of MQM.

It should be remarked that the background-independent theory does not mean that the physics is independent of the background. Background independence here means that, although a physical phenomenon occurs in a particular background with a specific initial condition, an underlying theory itself describing such a physical event should not presuppose any kind of spacetime or material backgrounds. Therefore, the background itself should arise from a vacuum solution of the underlying theory. In particular, the background-independent theory must integrate geometry and matter, as the matter cannot be defined without a preestablished spacetime framework. Complex spacetime structures are derived through the general deformations of the fundamental vacuum. These deformations correspond to physical processes that happen upon a particular (spacetime) background. Hence, they are regarded as a dynamical system. Motivated by a close analogy with quantum mechanics, we argued that the deformations of spacetime structure supported on a vacuum solution must be understood as the time evolution of the dynamical system. According to this picture, the fundamental action (76) describes a dynamical system, from which an emergent $(2n + 1)$-dimensional Lorentzian spacetime $\mathcal{M}$ with the metric (64) is derived.

Note that the Newton constant $G_N$ according to the emergent gravity picture has to be determined by field theory parameters only, such as the gauge coupling constant $G_{YM}$ and $\theta = B^{-1}$ defining the NC $U(1)$ gauge theory. In order to estimate the dynamical energy scale for the vacuum condensate (9), consider a simple dimensional analysis leading to the result [25,26]

$$\frac{G_N \hbar^2}{c^2} \sim G_{YM}^2 |\theta|, \tag{80}$$

where $|\theta| := |\mathrm{Pf}\theta|^{\frac{1}{n}}$. To be specific, when considering the four-dimensional case in which $M_P = (8\pi G_N)^{-1/2} \sim 10^{18}$ GeV and $G_{YM}^2 \sim \frac{1}{137}$, the vacuum energy (78) due to the condensate (9) is roughly given by

$$\rho_{\mathrm{vac}} = \frac{1}{4G_{YM}^2} |B_{ab}|^2 \sim G_{YM}^2 M_P^4 \sim 10^{-2} M_P^4. \tag{81}$$

Of course, its precise value may be given when the NC scale $|\theta| = M_{nc}^{-2}$ is known. In Equation (81), we roughly identified the NC scale $M_{nc}$ with the Planck energy $M_P$. However, this order of estimate is not so bad when we compare the value with that in Ref. [5] (see the last paragraph in Section III): $\rho_{\mathrm{vac}}^{1/4} = V_{60}^{1/4} = 3.8 \times 10^{16}$ GeV for 60 e-foldings. Then, the inflationary Hubble parameter corresponds to $H_{60} = 2.9 \times 10^{-5} M_P$. Emergent gravity reveals that the enigmatic vacuum energy $\rho_{\mathrm{vac}} \sim M_P^4$, rather surprisingly, serves as the true origin of flat spacetime. If spacetime geometry emerges from a vacuum configuration of some fundamental ingredients in an underlying quantum gravity theory, the Planck mass $M_P$ is a natural dynamical scale for the emergence of gravity and spacetime. Therefore, it may not be a surprising result but rather an inevitable consequence that the Planck energy density (81) in a vacuum was the genetic origin of spacetime.

The metric (64) clearly indicates that the Planck energy condensate in a vacuum resulted in an extremely extended spacetime. Since we have started with a background-independent theory in which any spacetime structure has not been assumed in advance, the spacetime was not existent at the beginning but simply emergent from the vacuum condensate (9). Therefore, the Planck energy condensation into a vacuum must be regarded as a dynamical process. Since the dynamical scale for the vacuum condensate is about that of the Planck energy, the time scale for the condensation will be roughly that of the Planck time $t_P \sim 10^{-44}$ s. The inflation scenario asserts that our Universe, at the beginning, underwent an explosive inflation era that lasted roughly $\sim 10^{-33}$ s. Thus, it is natural to consider the cosmic inflation as a dynamical process for the instantaneous condensation of vacuum energy to enormously spread out spacetime [23]. Now, we will explore how the cosmic inflation is triggered by the condensate of Planck energy in a vacuum and corresponds to the dynamical emergence of spacetime.

First, let us understand, intuitively, Equations (2) and (3) to obtain some clear insight from the old wisdom (see I.1 in [71]). Suppose that a test particle with mass $m$ is placed in the condensate with the energy density (81). Consider a ball of radius $r(t)$ and the test particle placed on its surface. According to Gauss's law, the particle will be subject to the gravitational potential energy $V(r) = -\frac{G_N M(r) m}{r}$ caused by the condensate (81), where $M(r) = \frac{4\pi r(t)^3 \rho_{\text{vac}}}{3}$ is the total mass inside the ball.[10] In order to preserve the total energy $E$ of the particle, the ball has to expand so that the kinetic energy $K(r) = \frac{1}{2} m \dot{r}(t)^2$ generated by the expansion compensates the negative potential energy. That is, the energy conservation implies the following relation

$$H^2 = \frac{8\pi G_N \rho_{\text{vac}}}{3} - \frac{k}{r(t)^2}, \tag{82}$$

where $H = \frac{\dot{r}(t)}{r(t)}$ is the expansion rate and $k \equiv -\frac{2E}{m}$. Comparing the above equation with the Friedmann Equation (2) after the identification $r(t) = Ra(t)$, we see that Equation (82) corresponds to $\rho_{\text{vac}} = V(\phi) \approx V_0$ and $\dot{\phi} \approx 0$ with $k = 0$. We actually assumed the spatially flat universe, $k = 0$, for the Friedmann Equation (2). In our approach, with a background-independent theory, the condition $k = 0$ is automatic since the very beginning should be absolutely nothing! This conclusion is consistent with the metric (64) which describes a final state of cosmic inflation. Hence, we may claim that the background-independent theory for cosmic inflation predicts a spatially flat universe, in which the constant $k$ must be exactly zero.

From the above simple argument (82) with $k = 0$, we see that the size of the ball exponentially expands, i.e.,

$$a(t) = a_0 e^{Ht} \tag{83}$$

where

$$H = \sqrt{\frac{8\pi G_N \rho_{\text{vac}}}{3}} \tag{84}$$

is a constant. Let us introduce fluctuations around the inflating solution (83) by considering $\rho_{\text{vac}} \to \rho_{\text{vac}} + \delta\rho$ and $\dot{\phi} \neq 0$, where $\delta\rho$ is the mechanical energy due to the fluctuations of the inflaton $\phi(t)$. Then Equation (82) is replaced by

$$H^2 = \frac{8\pi G_N}{3}(\rho_{\text{vac}} + \delta\rho), \tag{85}$$

and the dynamics of the inflaton is described by Equation (3). The argument leading to Equation (85) implies that the cosmic inflation corresponds to a dynamical process of the Planck energy condensation into a vacuum. Hence, the cosmic inflation as a dynamical system is typically a time-dependent solution and must be described by the non-Hamiltonian dynamics, as we already remarked in Equation (5). Now, we will demonstrate how the cosmic inflation can be described by the conformal Hamiltonian dynamics [72,73] which

appears in, for example, simple mechanical systems with friction. In Appendix A, we briefly review generalized symplectic manifolds that correspond to a natural phase space describing the conformal Hamiltonian dynamics.

Let us consider the simplest case, namely when the symplectic manifold is $\mathbb{R}^{2n}$ with coordinates $(q^i, p_i)$ and $\omega = dq^i \wedge dp_i = da$ where $a = \frac{1}{2}(q^i dp_i - p_i dq^i)$. A conformal vector field $X$ is defined by

$$\iota_X \omega = \kappa a + dH, \tag{86}$$

where $H : \mathbb{R}^{2n} \to \mathbb{R}$ is the Hamiltonian and $\kappa$ is a nonzero constant. Note that Equation (86) implies

$$\mathcal{L}_X \omega = \kappa \omega. \tag{87}$$

Therefore, the vector field $X$ is a Lie algebra generator of conformal infinitesimal transformations. It is easy to solve Equation (86) for the vector field $X$ and the result is given by

$$X = \frac{\kappa}{2}\left(q^i \frac{\partial}{\partial q^i} + p_i \frac{\partial}{\partial p_i}\right) + X_H, \tag{88}$$

where $X_H$ is a usual Hamiltonian vector field obeying $\iota_{X_H} \omega = dH$. Thus, Hamilton's equations are given by

$$\frac{dq^i}{dt} = X(q^i) = \frac{\kappa}{2}q^i + \frac{\partial H}{\partial p_i}, \tag{89}$$

$$\frac{dp_i}{dt} = X(p_i) = \frac{\kappa}{2}p_i - \frac{\partial H}{\partial q^i}. \tag{90}$$

The equations of motion for the Hamiltonian $H = \frac{1}{2}p_i^2 + U(q)$ are equal to the differential equations

$$\ddot{q}^i - \kappa \dot{q}^i + \frac{\partial V}{\partial q^i} = 0, \tag{91}$$

where $V(q) = U(q) + \frac{\kappa^2}{8}q_i^2$. To be specific, the integral curves for $U(q) = \frac{1}{2}\omega^2 q_i^2$ are given by[11]

$$q^i(t) = e^{\frac{\kappa}{2}t}q^i(\kappa = 0; t), \qquad p_i(t) = e^{\frac{\kappa}{2}t}p_i(\kappa = 0; t), \tag{92}$$

where $q^i(\kappa = 0; t) = A^i \sin(\omega t + \theta)$ and $p_i(\kappa = 0; t) = B_i \cos(\omega t + \theta)$ describe the usual harmonic oscillator with a closed orbit when $\kappa = 0$. The flow generated by a conformal vector field can be directly obtained by integrating Equation (87). Let $\phi_t$ denote the flow of $X$. By the Lie derivative theorem [61], we have $\frac{d}{dt}(\phi_t^* \omega) = \phi_t^* \mathcal{L}_X \omega = \kappa \phi_t^* \omega$. Therefore, we see that the conformal flow has the property

$$\phi^* \omega = e^{\kappa t} \omega. \tag{93}$$

This means that the volume of phase space exponentially expands (contracts) if $\kappa > 0$ ($\kappa < 0$).

The mathematical parallel between quantum mechanics and NC spacetime offers insights into formulating cosmic inflation as a dynamical system. First note that the NC space (19) in commutative limit becomes a phase space with the symplectic form

$$B = \frac{1}{2}B_{\mu\nu}dy^\mu \wedge dy^\nu. \tag{94}$$

Hamiltonian systems generated by divergenceless Hamiltonian flows are characterized by the invariance of phase space volume under time evolution, which is known as the Liouville theorem [60,61]. However, the cosmic inflation indicates that the volume of spacetime phase space has to exponentially expand as seen from the above mechanical analogue. Hence, a generalized Liouville theorem is necessary to describe the exponential expansion of spacetime. We have already observed how a non-Hamiltonian dynamics

can be formulated in terms of a *conformal* Hamiltonian dynamics characterized by the flow obeying Equation (87). See Appendix A for a mathematical exposition of general time-dependent nonconservative dynamical systems.

Let us apply the conformal Hamiltonian dynamics to the cosmic inflation. Recall that we have considered an atlas $\{(U_i, \phi_{(i)})\}$ on $M = \bigcup_{i \in I} U_i$ as a collection of local Darboux charts and complete it by gluing these local charts on their overlap. On each local chart, we have a local symplectic structure $\Omega_i = \frac{1}{2} B_{\mu\nu} dy^\mu_{(i)} \wedge dy^\nu_{(i)}$ where $\{y^\mu_{(i)}\}$ are Darboux coordinates on a local patch $U_i \subset M$. The phase space coordinates $\{y^\mu_{(i)}\}_{U_i}$ of a conformal Hamiltonian system undergo a nontrivial time evolution even in a local Darboux frame [74,75]. For example, the equations of motion (89) and (90) illustrate such a nontrivial time evolution even when $H = 0$. The dynamics in this case consists of the orbits of a conformal vector field $X$ obeying the condition (87). The situation at hand is essentially the same as the mechanical system with negative friction. To be specific, write $\Omega_i = da_{(i)}$ on a local patch $U_i \subset M$ where $a_{(i)} = -\frac{1}{2} p^{(i)}_\mu dy^\mu_{(i)}$ and $p^{(i)}_\mu = B_{\mu\nu} y^\nu_{(i)}$. Define a conformal vector field $X$ as

$$\iota_X \Omega_i = \kappa a_{(i)} + dH_i, \tag{95}$$

where $H_i : U_i \to \mathbb{R}$ is a local Hamiltonian and $\kappa$ is a positive constant. Using the fact that $d\Omega_i = 0$, Equation (95) can be written as

$$\mathcal{L}_X \Omega_i = \kappa \Omega_i. \tag{96}$$

The vector field $X$ obeying Equation (95) is given by

$$X = \frac{\kappa}{2} y^\mu_{(i)} \frac{\partial}{\partial y^\mu_{(i)}} + X_{H_i}, \tag{97}$$

where $X_{H_i}$ is the ordinary Hamiltonian vector field satisfying $\iota(X_{H_i})\Omega_i = dH_i$. The conformal vector field (97) contains the Liouville vector field $Z_{(i)} \equiv \frac{1}{2} y^\mu_{(i)} \frac{\partial}{\partial y^\mu_{(i)}}$ [72,73].

Let us consider a spacetime dynamics generated by the Liouville vector field. We will set $H_i = 0$ for simplicity. The time evolution of local Darboux coordinates is then determined by the equations

$$\frac{dy^\mu_{(i)}}{dt} = X(y^\mu_{(i)}) = \frac{\kappa}{2} y^\mu_{(i)}. \tag{98}$$

The solution is given by

$$y^\mu_{(i)}(t) = e^{\frac{\kappa}{2}t} y^\mu_{(i)}(0). \tag{99}$$

We may glue the local solutions (99) to have a global form

$$p_a(t) = B_{ab} y^b(t) = e^{\frac{\kappa}{2}t} p_a. \tag{100}$$

Then the time-dependent canonical one-form is given by

$$a(t) = -\frac{1}{2} p_a(t) dy^a(t) = -\frac{1}{2} e^{\kappa t} p_a dy^a \tag{101}$$

and thus

$$\Omega(t) = da(t) = e^{\kappa t} B. \tag{102}$$

The exterior derivative above acts only on $\mathbb{R}^{2n}$. One can show that the result (102) is the integral form of Equation (96). More generally, the result (102) is a particular case of the

general Moser flow $\phi_t$ generated by a time-dependent vector field $X_t$ for a locally conformal symplectic manifold [76]. The Moser flow satisfies

$$\phi_t^* \Omega_t = \exp\left(\int_0^t \phi_s^*(b_s(X_s)) ds\right) \cdot \Omega, \tag{103}$$

where the one-form $b$ is the Lee form of $\Omega$ [77]. The result (102) is simply obtained from Equation (103) when $b(X)$ is a constant $\kappa$.

We have advanced the concept of cosmic inflation by postulating that the vacuum configuration (71) serves as a final state accumulating the vacuum energy. Therefore, the cosmic inflation corresponds to a dynamical system describing the transition from the initial state referring to "absolutely nothing" to the final state. With this perspective in mind, let us consider a symplectic manifold $(M, \Omega(t))$ whose symplectic two-form is given by Equation (102). It can be shown that this symplectic manifold arises from a time-dependent vacuum given by

$$\langle \phi_a(t) \rangle_{\text{vac}} = p_a(t) = e^{\frac{\kappa}{2}t} p_a, \qquad \langle A_0(t) \rangle_{\text{vac}} = 0. \tag{104}$$

Recall that the temporal gauge field in Equation (104) corresponds to our previous setting $H_i = 0$ according to the identification (43). Though we have turned off the temporal gauge field for a simple argument, it is necessary to turn it on in order to implement the vacuum (104) as a solution of the action (76). We will consider it later. Let us first determine the vacuum geometry emergent from the vacuum configuration (104). Note that

$$\langle [\phi_a(t), \phi_b(t)] \rangle_{\text{vac}} = -i e^{\kappa t} B_{ab} = -i \Omega_{ab}(t), \tag{105}$$

and so we regard $\Omega(t) = \frac{1}{2} \Omega_{ab}(t) dy^a \wedge dy^b$ as the symplectic structure of the inflating vacuum (104). The vacuum (104) leads to the vector fields (omitting the symbol indicating the vacuum for a notational simplicity)

$$V_A(t) = (V_0(t), V_a(t)) = \left(\frac{\partial}{\partial t}, e^{\frac{\kappa}{2}t} V_a(0)\right), \tag{106}$$

where $V_a(0) = \delta_a^\mu \frac{\partial}{\partial y^\mu}$. Thus, the dual one-forms are given by

$$v^0(t) = dt, \qquad v^a(t) = e^{-\frac{\kappa}{2}t} v^a(0) \tag{107}$$

where $v^a(0) = \delta_\mu^a dy^\mu$. It is easy to calculate the Lie algebra (62) for the time-dependent vector fields $V_A(t)$:

$$g_{AB}{}^C = \begin{cases} g_{0a}{}^b = -g_{a0}{}^b = -\frac{\kappa}{2}\delta_a^b, & a, b = 1, \cdots, 2n; \\ 0, & \text{otherwise.} \end{cases} \tag{108}$$

Thus, $\lambda^2 = e^{n\kappa t}$ according to Equation (63). The invariant volume form of the vacuum manifold is then given by

$$\nu_t = \lambda^2 dt \wedge v^1(t) \wedge \cdots \wedge v^{2n}(t) = dt \wedge dy^1 \wedge \cdots \wedge dy^{2n}. \tag{109}$$

After applying the above results to the metric (64), we see that the vacuum configuration (104) determines the spacetime geometry with the metric

$$ds^2 = -dt^2 + e^{2Ht} d\mathbf{y} \cdot d\mathbf{y} \tag{110}$$

where $H = \frac{1}{2}(n-1)\kappa > 0$ is a positive Hubble parameter. This is the de Sitter space in flat coordinates which covers half of the de Sitter manifold. Definitely, the inflation metric (110) describes a homogeneous and isotropic Universe known as the Friedmann–Robertson–Walker metric in physical cosmology.

The vector fields $V_A(t)$ in Equation (108) form a solvable Lie algebra and the de Sitter space is its Lie group. The Lie algebra has the generators $V_0 = -\frac{\kappa}{2}L_{0(2n+1)}$, $V_a = \frac{1}{2}(L_{0a} + L_{a(2n+1)})$, which is indeed a subalgebra of the de Sitter algebra where $L_{\mathbb{A}\mathbb{B}}$ ($\mathbb{A}, \mathbb{B} = 0, 1, \cdots, 2n+1$) are the Lie algebra generators of $SO(2n+1, 1)$ Lorentz symmetry. From this point of view, energy and momentum do not commute, unlike in the Minkowski spacetime, and are no longer conserved, as translations are no longer a symmetry of the space.[12] Instead, energy generates scale transformations in momentum. This is the reason why the isometry of the de Sitter space is enhanced to $SO(2n+1, 1)$ which combines $SO(2n, 1)$ Lorentz transformations and translations together [80]. In the limit $\kappa \to 0$, we recover the Minkowski spacetime.

In order to achieve a background-independent formulation of emergent spacetime, it is desirable to realize the inflationary universe as a solution of the matrix model (76). Now, we will show that the cosmic inflation arises as a time-dependent solution describing the dynamical process of Planck energy condensate into a vacuum without introducing any inflaton field as well as an ad hoc inflation potential. First, let us show that the dynamical process for the vacuum condensate is described by the time-dependent vacuum configuration given by

$$\langle \phi_a(t) \rangle_{\text{vac}} = p_a(t) = e^{\frac{\kappa t}{2}} p_a, \qquad \langle \widehat{A}_0(t) \rangle_{\text{vac}} = \widehat{a}_0(t, y), \tag{111}$$

where the temporal gauge field is given by an open Wilson line [81–83]

$$\widehat{a}_0(t, y) = \frac{\kappa}{2} \int_0^1 d\sigma \frac{dy^a(\sigma)}{d\sigma} p_a(\sigma) \tag{112}$$

along a path parameterized by the curve $y^a(\sigma) = y_0^a + \zeta^a(\sigma)$ where $\zeta^a(\sigma) = \theta^{ab}k_b\sigma$ with $0 \le \sigma \le 1$ and $y^a(\sigma = 0) \equiv y_0^a$ and $y^a(\sigma = 1) \equiv y^a$. The constant $\kappa$ will be identified with the inflationary Hubble constant $H$. Note that the second term in Equation (69) identically vanishes for the background (111). Therefore, it is enough to impose the condition

$$D_0\phi_a = e^{\frac{\kappa t}{2}} \left( \frac{\kappa}{2}p_a - i[\widehat{A}_0, p_a] \right) = 0 \tag{113}$$

to satisfy both (69) and (70). In terms of the NC $\star$-algebra $\mathcal{A}_\theta^1$, Equation (113) reads as

$$\frac{\partial \widehat{a}_0(t, y)}{\partial y^a} = \frac{\kappa}{2}p_a. \tag{114}$$

Using the formula

$$\frac{\partial}{\partial y^a} \int_0^1 d\sigma \frac{dy^b(\sigma)}{d\sigma} K(y(\sigma)) = \delta_a^b K(y) \tag{115}$$

for some differentiable function $K(y)$, one can easily check that the temporal gauge field in Equation (112) satisfies Equation (114).

We want to address some physical significance of the nonlocal term (112). It is essential to highlight that the temporal gauge field (112) corresponds to a background Hamiltonian density in the comoving frame.[13] It will be shown that, although the temporal gauge field (112) is nonlocal, the gravitational metric determined by the time-dependent vacuum configuration (111) still takes a local expression as it should be. It was already noticed in [84] that nonlocal observables in emergent gravity are, in general, necessary to describe some gravitational metric. Indeed the appearance of such nonlocal terms should not be surprising since there exist no local gauge invariant observables in NC gauge theories [81–83].

Now, let us determine the metric (64) for the inflating background (111). The $(2n+1)$-dimensional vector fields defined by Equation (53) take the following form

$$V_0(t) = \frac{\partial}{\partial t} - \frac{\kappa}{2}y^a\frac{\partial}{\partial y^a}, \qquad V_a(t) = e^{\frac{\kappa t}{2}}\frac{\partial}{\partial y^a}. \tag{116}$$

Higher-order derivative terms in Equations (54) and (55) identically vanish since only the vacuum background (111) was considered. Note that the vector fields take the local form again as the result of applying Equation (115) and the open Wilson line (112) leads to a conformal vector field $Z \equiv \frac{1}{2} y^a \frac{\partial}{\partial y^a}$ known as the Liouville vector field [72,73]. Then, the dual orthogonal one-forms are given by

$$v^0(t) = dt, \qquad v^a(t) = e^{-\frac{\kappa t}{2}}(dy^a + \mathbf{a}^a) = e^{-\kappa t} dy_t^a \qquad (117)$$

where

$$\mathbf{a}^a = \frac{\kappa}{2} y^a dt, \qquad y_t^a \equiv e^{\frac{\kappa t}{2}} y^a. \qquad (118)$$

One can see that the vector fields in Equation (116) satisfy $[V_0, V_a] = \kappa V_a$, and thus

$$g_{AB}{}^C = \begin{cases} g_{0a}{}^b = -g_{a0}{}^b = -\kappa \delta_a^b, & a, b = 1, \cdots, 2n; \\ 0, & \text{otherwise.} \end{cases} \qquad (119)$$

From this result, we obtain $\lambda^2 = e^{2n\kappa t}$ since $g_{BA}{}^B = V_A \ln \lambda^2$ [26,39]. One can see that the volume-preserving condition (60) is satisfied since $\rho = e^{n\kappa t}$, $A_0^a = -\frac{\kappa}{2} y^a$ and $V_a^\mu = e^{\frac{\kappa t}{2}} \delta_a^\mu$. In the end, the time-dependent metric for the inflating background (111) is given by

$$ds^2 = -dt^2 + e^{2Ht} d\mathbf{y}_t \cdot d\mathbf{y}_t, \qquad (120)$$

where we have identified the inflationary Hubble constant $H \equiv (n-1)\kappa$. By comparing this result with Equation (110), one can see that the temporal gauge field (112) enhances the inflation by the factor two, i.e., $H \to 2H$. We emphasize that the temporal gauge field (112) is crucial to satisfy Equations (69) and (70).

We demonstrated that cosmic inflation arises as a time-dependent solution of a background-independent theory. This theory delineates the dynamical evolution of the Planck energy condensate in a vacuum, without introducing an inflaton field or an ad hoc inflation potential. Let us generalize the cosmic inflation by also including arbitrary fluctuations around the inflationary background (111). Such a general inflationary universe in $(2n + 1)$-dimensional Lorentzian spacetime can be realized by considering a time-dependent NC algebra given by

$$^t\mathcal{A}_\theta^1 \equiv \left\{ \widehat{\phi}_0(t, y) = i \frac{\partial}{\partial t} + \widehat{A}_0(t, y), \quad \widehat{\phi}_a(t, y) = e^{\frac{\kappa t}{2}} (p_a + \widehat{A}_a(t, y)) \right\}. \qquad (121)$$

We denote the corresponding time-dependent matrix algebra by $^t\mathcal{A}_N^1$ which consists of time-dependent solutions of the action (76). Then, the general Lorentzian metric describing a $(2n + 1)$-dimensional inflationary universe can be obtained by the following duality chain:

$$^t\mathcal{A}_N^1 \quad \Longrightarrow \quad ^t\mathcal{A}_\theta^1 \quad \Longrightarrow \quad ^t\mathfrak{D}^1. \qquad (122)$$

The module $^t\mathfrak{D}^1$ of derivations of the NC algebra $^t\mathcal{A}_\theta^1$ is given by

$$^t\mathfrak{D}^1 = \left\{ \widehat{V}_A(t) = (\widehat{V}_0, \widehat{V}_a)(t) | \widehat{V}_0(t) = \frac{\partial}{\partial t} + \mathrm{ad}_{\widehat{A}_0(t,y)}, \quad \widehat{V}_a(t) = e^{\frac{\kappa t}{2}} \left( \frac{\partial}{\partial y^a} + \mathrm{ad}_{\widehat{A}_a(t,y)} \right) \right\}, \qquad (123)$$

where the adjoint operations are defined by Equation (23). In the classical limit of the module (123), we obtain a general inflationary universe described by

$$ds^2 = -dt^2 + e^{2Ht}(1 + \delta\lambda)^2 v_b^a v_c^a (dy_t^b - \mathbf{A}^b)(dy_t^c - \mathbf{A}^c), \qquad (124)$$

where $v_b^a := v_b^a(t, y)$, $\delta\lambda := \delta\lambda(t, y)$ and $\mathbf{A}^b := \delta a_0^b(t, y) dt$. If all fluctuations are turned off for which $v_b^a = \delta_b^a$ and $\delta\lambda = \mathbf{A}^b = 0$, we recover the inflation metric (120).

Let us bring this section to a close by delving into the physical implications arising from the results we have garnered. Recall that an NC space such as $\mathbb{R}_\theta^2$ does not admit a state

defined on a single point of the plane but, rather, the state lies in a region of the plane. Thus, there must be a basic length scale, below which the notion of space (and time) does not make sense. Let us fix such a typical length scale at $t = 0$ as $|y^a(t = 0)| \sim L_P$ or $l_s = \sqrt{\alpha'}$ using the scaling freedom noted in footnote 9. Since we have started with a background-independent theory in which a spacetime structure has to be created from a solution at the beginning $t = 0$, it should be reasonable to identify $L_P$ with the Planck length. Since $y^a(t = 0)$ are operators acting on a Hilbert space, this means that the inflationary vacuum (111) creates a spacetime of the Planck size. After the creation, the universe undergoes the inflation epoch described by a solution of the time-dependent matrix model, unlike the traditional inflationary models that suppose just the exponential expansion of a preexisting spacetime. This picture is similar to the birth of inflationary universes in Refs. [12,13] in which the universe is spontaneously created by quantum tunneling from nothing into a de Sitter space. Here, "nothing" means a state devoid of any spacetime structure. According to the standard inflation scenario, the universe expanded by at least a factor of $e^{60} \sim 10^{26}$ during the inflation. After 60 e-foldings at $t = t_{\text{end}} = 10^{-36} \sim 10^{-33}$ s, $Ht_{\text{end}} \gtrsim 60$ and the size of the universe at the end of inflation amounts to $|y^a(t = t_{\text{end}})| = e^{Ht_{\text{end}}}|y^a(t = 0)| \gtrsim 10^{26}L_P$. Since $1 \, \text{eV} = (6.6 \times 10^{-16} \, \text{s})^{-1}$, this roughly informs us of the energy scale of the inflationary Hubble constant $H \gtrsim 10^{11} \sim 10^{14}$ GeV [3–5].

Since the vacuum (111) is in high nonequilibrium (i.e., time-dependent), it is expected that it undergoes evolutionary processes towards its final state (71) through interactions with its environment, such as ubiquitous fluctuations. This dissipation process of inflation energy is known as the reheating mechanism in physical cosmology. To accurately ascertain the duration of inflation, the precise mechanism involved in reheating must be understood; unfortunately, this surpasses our current knowledge. Nevertheless, we will speculate in Section 4 about a plausible picture for the reheating mechanism.

## 4. Discussion

String theory has been developed upon two distinct spacetime frameworks, namely the Kaluza–Klein (KK) theory and emergent gravity. Despite their conceptual divergence, these models represent exclusive perspectives on the nature of spacetime. On the one hand, KK gravity is defined in higher dimensions as a more superordinate theory and gauge theories in lower dimensions are derived from the KK theory via compactification. Since the KK theory is just Einstein gravity in higher dimensions, the prior existence of spacetime is a priori assumed. On the other hand, in the emergent gravity picture, gravity in higher dimensions is not a fundamental force but a collective phenomenon emergent from more fundamental ingredients defined in lower dimensions. In the emergent gravity approach, the existence of spacetime is not a priori assumed, but the spacetime structure is defined by the theory itself. This picture leads to the concept of emergent spacetime. In some sense, emergent gravity is the inverse of the KK paradigm, schematically summarized by [24]

$$(1 \otimes 1)_S \rightleftarrows 2 \oplus 0 \tag{125}$$

where $\rightarrow$ means the emergent gravity picture, while $\leftarrow$ indicates the KK picture.

Recent developments in string theory have revealed growing evidence for emergent gravity and emergent spacetime. The AdS/CFT correspondence and matrix models are typical examples supporting the emergence of gravity and spacetime [14–16]. An intriguing aspect is that the emergence of gravity requires the emergence of spacetime too. If spacetime is emergent, everything supported on the spacetime should be emergent too, ensuring internal consistency within the theoretical framework. In particular, matters cannot exist without spacetime, and thus, must be emergent together with the spacetime. Eventually, the background-independent theory has to make no distinction between geometry and matter [27]. This is the reason why the emergent spacetime picture cannot coexist peacefully with the KK paradigm. Since the emergent spacetime is a new fundamental paradigm for quantum gravity and radically different from any previous physical theories, all of which describe what happens in a given spacetime, there is a compelling need to critically

reassess the underpinnings of quantum gravity through the lens of emergent spacetime. Quantum gravity is considered necessary for a complete understanding of cosmic inflation because inflationary theory involves the extreme conditions of the early universe where both quantum mechanics and gravity play significant roles.

It is well known [66,85–88] that NC field theories arise as a low-energy effective theory in string theory, in particular, on D-branes upon turning on a constant $B$-field. A remarkable aspect of the NC field theory is that it can be mapped to a large $N$ matrix model as depicted in the isomorphism (22). The relation between NC gauge theories and matrix models is quite general since any Lie algebra or Moyal-type NC space such as (19) always admits a separable Hilbert space, and NC gauge fields become operators acting on the Hilbert space [57]. The matrix representation of NC gauge fields implies that they can be embedded into a background-independent formulation in terms of a matrix model. The background-independent variables are identified as the degrees of freedom inherent in the underlying matrix model. The relation between a matrix model and an NC gauge theory is based on the observation [54,57] that the NC space (19) is a consistent vacuum solution of a large $N$ gauge theory in the Coulomb branch. The matrices are original dynamical variables of the matrix model which are manifestly background-independent and NC gauge fields are derived from fluctuations in the NC Coulomb branch.

We have shown that the cosmic inflation arises as a solution of a time-dependent matrix model, describing the dynamical process of the vacuum energy condensation. Remarkably, the inflation can be described by time-dependent matrices only without introducing any inflaton field as well as an ad hoc inflation potential. In order to describe the cosmic inflation, it is necessary to generalize symplectic manifolds, as we have discussed the rationales in Section 3. The corresponding generalized symplectic manifolds for the cosmic inflation include locally conformal symplectic (LCS) or more generally locally conformal cosymplectic (LCC) manifolds, whose mathematical foundation will be reviewed in Appendix A. The LCS manifold allows a nontrivial conformal vector field defined by Equation (96) even when an underlying Hamiltonian function identically vanishes. The so-called Liouville vector field $Z \equiv \frac{1}{2} y^\mu \frac{\partial}{\partial y^\mu}$ is still nontrivial [72] and it generates the exponential expansion of spacetime described by the metric (110).[14] If the one-form $a$ in Equation (95) is proportional to the Lee form $b$, $X$ is called a Hamiltonian vector field of an LCS manifold. See the definition (A10). The Hamiltonian vector field in this case shows a peculiar property different from the symplectic case: If $b$ is not exact, $X_H = 0$ only if $H = 0$ (see Proposition 2.1 in [74]). Therefore, we see that the vector fields of an LCS manifold are in stark contrast to those of a symplectic manifold, in which $X_H = 0$ if and only if $H = $ constant. Due to this property, while the constant vacuum energy (i.e., a cosmological constant) does not couple to gravity if gravity is described by a symplectic manifold, the vacuum energy rightly couples to gravity during the inflation if the cosmic inflation is described by an LCS (or more generally LCC) manifold. This is a desirable property since the cosmic inflation is triggered by the condensate of vacuum energy. Physically, the reason is obvious since all quantities during the inflation are time-dependent due to the existence of the nontrivial Liouville vector field.

It may be instructive to understand the above situation more closely in comparison with the equilibrium case described by the metric (64). First, note that the invariant volume form (61) can be written as

$$\nu_t = \lambda^{2-2n} \nu_g, \tag{126}$$

where $\nu_g = e^0 \wedge \cdots \wedge e^{2n} = \sqrt{-\mathcal{G}} d^{2n+1} x$ is the volume form of the metric. Therefore, the vector fields $V_A$ do not necessarily preserve the Riemannian volume form $\nu_g$ although they preserve the volume form $\nu_t$. However, since $\lambda^2 \to 1$ at spatial infinity according to Equation (63), $\nu_t|_\infty = \nu_g|_\infty$ for the asymptotic volume forms denoted by $\nu_t|_\infty$ and $\nu_g|_\infty$. Therefore, the flow generated by $V_A$ leads to only local changes in the spacetime volume, while it preserves the volume element at asymptotic regions. Conversely, the conformal vector field (97) changes the spacetime volume everywhere. Accordingly, it definitely gives rise to the exponential expansion of the spacetime volume. After all, we see that a natural

phase space for the cosmic inflation has to contain an LCS manifold instead of a standard symplectic manifold. Including time, it becomes an LCC manifold [75]. Our result shows that the matrix model (68) contains the LCC manifold as a solution.

An important question is whether the emergent spacetime picture can also lead to the eternal (or chaotic) inflation. The answer is certainly no. The reason is the following. We showed that the inflationary vacuum (111) arises as a solution of the (BFSS-like) matrix model (76). In order to define the matrix model (76), however, we have not introduced any spacetime structure. Hence, the vacuum (111) corresponds to the creation of spacetime unlike the traditional inflationary models that describe just the exponential expansion of a preexisting spacetime. More precisely, the inflationary vacuum (111) describes a dynamical process of the Planck energy condensate responsible for the emergence of spacetime. In general relativity, the Minkowski spacetime with the metric $g_{\mu\nu} = \eta_{\mu\nu}$ must be a completely empty space because the Einstein Equation (1) requires $T_{\mu\nu} = 0$. However, in emergent gravity, it is not an empty space but the vacuum condensate of the Planck energy as Equation (81) clearly indicates. An important point is that the Planck energy condensate results in a highly coherent vacuum called the NC space, and the NC space is identical to the NC phase space in quantum mechanics which necessarily brings about the Heisenberg's uncertainty relation, $\Delta x \Delta p \geq \frac{\hbar}{2}$. Thus, the NC space (19) also leads to the spacetime uncertainty relation. Therefore, any further accumulation of energy over the vacuum (111) must be subject to the spacetime exclusion principle known as the UV/IR mixing [89]. Consequently, it is not possible to further accumulate the Planck energy density over the inflationary vacuum (111). This means that it is impossible to superpose a new inflating subregion over the inflationary vacuum. Rather, it was argued [90] that the UV/IR mixing due to the spacetime uncertainty principle gives rise to a late-time acceleration of the universe, also known as the dark energy.

In sum, the cosmic inflation triggered by the Planck energy condensate into a vacuum must be a single event [23] and the emergent spacetime precludes the formation of pocket universes appearing in the eternal (or chaotic) inflation. In the end, we have a beautiful picture: the NC spacetime is necessary for the emergence of spacetime and the exclusion principle of NC spacetime guarantees the stability of spacetime.

We certainly live in a universe where the inflationary epoch lasted for only a very tiny period in very early times, although it is currently in an accelerating phase driven by dark energy. Therefore, there should be some relaxation mechanism for the (first-order) phase transition from the inflating universe to a radiation-dominated universe. We showed that the former is described by the metric (124), whereas the latter is described by (64), and both arise as solutions of the background-independent matrix model (68). In inflation scenarios in terms of scalar fields, the relaxation mechanism is known as the reheating in which the scalar field switches from being overdamped to being underdamped and begins to oscillate at the bottom of the potential to transfer its energy to a radiation-dominated plasma at a sufficiently high temperature to allow standard big bang nucleosynthesis [3,4]. For this purpose, most inflationary theories have introduced a very ad hoc potential for the scalar field (inflaton). In our case, however, we have introduced neither an inflaton field nor an inflation potential. Therefore, the important question is how to end the inflation of our universe in the emergent gravity.

We do not know the precise mechanism for the graceful exit. Thereby, we will briefly speculate a plausible scenario only. Let us start with a naive observation. The Lorentzian metric (124) describes general scalar–tensor perturbations on the inflating spacetime. Since the fluctuations have been superposed on the inflating background, we suspect that there may be some nonlinear damping mechanism through the interactions between the background and the density fluctuations. To be precise, there may be a cosmic analogue of the Landau damping in plasma physics originally applied to longitudinal oscillations of an electron plasma. The Landau damping in plasma occurs due to the energy exchange between an electromagnetic wave and particles in the plasma with a velocity approximately equal to the phase velocity of the wave. It leads to exponentially decaying collective oscilla-

tions.[15] The Landau damping may be intuitively understood by considering how a surfer gains energy from a sea wave. For the wave to be damped, the wave velocity and the surfer velocity must be similar, and then the surfer is trapped by the wave. If the surfer is slightly slower than the wave mode, the mode loses energy compared to the surfer. A similar situation may happen in the inflating spacetime (124). Local fluctuations (cf., surfers) on the inflating spacetime (cf., the wave mode) are given by Equation (121). Note that these local fluctuations carry an additional localized energy and this local energy will cause a slight delay in the drift of local lumps compared to the inflating background. Moreover, these drift delays will occur everywhere since (quantum) fluctuations are ubiquitous. Then, this is precisely the condition for the Landau damping to occur. If this is true, the inflating mode will transfer its inflation (potential) energy to ubiquitous local fluctuations, ending the inflation through an exponential damping and entering into a radiation-dominated era via the reheating at a sufficiently high temperature for the standard Big Bang.

The above speculation may not be so absurd, considering the fact that the cosmic inflation is described by a conformal Hamiltonian system [72,73] which also appears in dynamical systems with friction and the transition of such dynamical systems in nonequilibrium into equilibrium is induced by interactions with environment. For the cosmic inflation, ubiquitous fluctuations over the inflating spacetime will play a role in the environment. This speculation may be further supported by the fact that the underlying theory for emergent gravity is Maxwell's electromagnetism on NC spacetime and the Landau damping can be realized even at a nonlinear level [91]. Therefore, it will be interesting to verify whether the naive idea can work or not. Probably, the cosmic Landau damping may be closely related to the instability of de Sitter space as suggested by Polyakov [78,79].

Our real world, $\mathbb{R}^{1,3} \cong \mathbb{R} \times \mathbb{R}^3$, is as mystic as ever because the spatial 3-manifold does not belong to the family of (almost) symplectic manifolds. Let us enumerate potential pathways leading to our tangible reality—the four-dimensional Lorentzian spacetime $\mathcal{M}$:

A.   Analytic continuation or Wick rotation from $\mathbb{R}^4$.
B.   Kaluza–Klein compactification $\mathcal{M} \times \mathbb{S}^1$.
C.   Contact manifold $(\mathbb{R}^3, \eta)$.
D.   Nambu structure $(\mathbb{R}^3, C)$.

Here, $\eta = dz - ydx$ is a contact form on $\mathbb{R}^3$ and $C = \frac{1}{3!}C_{\mu\nu\lambda}dx^\mu \wedge dx^\nu \wedge dx^\lambda$ is a nondegenerate, closed three-form on $\mathbb{R}^3$. In case (A), the Lorentzian metric is obtained from Equation (33) with $n = 2$ by the Wick rotation $y^4 = iy^0$. It is also straightforward to compactify the $(4 + 1)$-dimensional Lorentzian metric (64) onto $\mathbb{S}^1$ to obtain the result (B). Since the time is also defined as a contact structure, case (C) has two contact structures as with the matrix string theory discussed in Appendix C. It may be interesting to briefly explore some clues for the cosmic inflation in context (C). Let $N = \mathbb{R} \times \mathbb{R}^3$ and $t \in \mathbb{R}$ be the time coordinate and $f_t = f(t)$ be a positive monotonic function. Define a time-dependent closed two-form on $N$ by

$$B_t = d\lambda_t = f_t(dT \wedge \eta + d\eta) \tag{127}$$

where $\lambda_t = f_t\eta$ and $T = \ln f_t$. Since $B_t^2 = 2e^{2T}dT \wedge \eta \wedge d\eta$ is vanishing nowhere, $B_t$ is a symplectic structure on $N$. Consider a time-dependent Hamiltonian $H : N \rightarrow \mathbb{R}$ such that $dH = -e^T dT$ and denote the Hamiltonian vector field of $H$ by $X_H$. Let $R$ be the Reeb vector field associated with the contact form $\eta$ (see Appendix A for the definition). Then, it is easy to show that

$$\iota_R B_t = dH, \tag{128}$$

that is, $R = X_H$. A very interesting property is that

$$Z = \frac{\partial}{\partial T} \tag{129}$$

is the Liouville vector field of the symplectic form $B_t$, i.e., $\mathcal{L}_Z B_t = B_t$ or $\iota_Z B_t = \lambda_t$. This condition can be written as $\mathcal{L}_Z \lambda_t = \lambda_t$. One can regard the Liouville vector field $Z$ as

the Reeb vector field associated with the contact form $dT$. Since $\iota_Z(B_t^2) = 2e^{2T}\eta \wedge d\eta$, the one-form $\lambda_t$ gives rise to a contact form on every three-dimensional submanifold $M \subset N$ transverse to $Z$. Thus, we expect that the conformal vector field $Z$ will generate an inflationary metric given by

$$ds^2 = -dT^2 + e^{2T}d\mathbf{x} \cdot d\mathbf{x}. \tag{130}$$

It will be interesting to have a microscopic derivation of the above inflation metric from the matrix string theory (A79). The approach in [43] may be useful for this case. Given our current lack of understanding in formulating emergent gravity based on the Nambu structure (D), the realization of this concept remains a distant aspiration. It may be of M-theory origin because it is involved with the 3-form $C$ instead of the symplectic 2-form $B$.

**Funding:** This work was supported by the National Research Foundation of Korea (NRF) with grant number NRF-2018R1D1A1B0705011314.

**Data Availability Statement:** No new data were created or analyzed in this study. Data sharing is not applicable to this article.

**Acknowledgments:** We would like to thank Seokcheon Lee for their helpful discussions and Jungjai Lee for their insightful discussions on the Landau damping.

**Conflicts of Interest:** The authors declare no conflict of interest.

## Appendix A. Locally Conformal Cosymplectic Manifolds

In this Appendix, we briefly review locally conformal cosymplectic (LCC) manifolds. It was shown in [75] that an LCC manifold can be seen as a generalized phase space of a time-dependent Hamiltonian system. We will apply the results in Refs. [74,75] to emergent gravity and argue that the LCC manifold is a natural phase space describing the cosmic inflation of our universe.

First, let us consider locally conformal symplectic (LCS) manifolds. An LCS manifold is a triple $(M, \Omega, b)$ where $b$ is a closed one-form and $\Omega$ is a nondegenerate (but not closed) two-form satisfying

$$d\Omega - b \wedge \Omega = 0. \tag{A1}$$

The dimension of $M$ will be assumed to be at least 4 and the one-form $b$ is called the Lee form [77]. A symplectic manifold corresponds to the case with $b = 0$. If the Lee form $b$ is exact, the manifold is globally conformal symplectic (GCS). Locally, by choosing $b = d\lambda^{(\alpha)}$ for a local function $\lambda^{(\alpha)} : U_\alpha \to \mathbb{R}$ on an open neighborhood $U_\alpha$, Equation (A1) is equivalent to $d(e^{-\lambda^{(\alpha)}}\Omega) = 0$, so the local geometry of LCS manifolds is exactly the same as that of symplectic manifolds. Thus, an LCS form on a manifold $M$ is a nondegenerate two-form $\Omega$ that is locally conformal to a symplectic form. In other words, on an LCS manifold $(M, \Omega, b)$, there exists an open covering $\{U_\alpha\}$ of $M$ and a smooth positive function $f_\alpha$ on each $U_\alpha$ such that $f_\alpha\Omega|_{U_\alpha}$ is symplectic on $U_\alpha$. Two LCS forms $\Omega$ and $\Omega'$ are said to be (conformally) equivalent if there exists some positive function $f$ such that $\Omega' = f\Omega$, where the Lee form of $\Omega'$ is just $b' = b + d\ln f$. An interesting example [92] is the Hopf manifolds that are diffeomorphic to $\mathbb{S}^1 \times \mathbb{S}^{2n-1}$ and have a locally conformal Kähler metric, while they admit no Kähler metric.

An LCS manifold can be seen as a generalized phase space of Hamiltonian dynamical systems since the form of Hamilton's equations is preserved by homothetic canonical transformations. Let us recapitulate how the LCS manifolds naturally arise from the Hamiltonian dynamics of particles. Consider a dynamical system with $n$ degrees of freedom so that its phase space is a $2n$-dimensional differentiable manifold $M$ endowed with an open covering of coordinate neighborhoods $\{U_\alpha\}_{\alpha \in I}$ with local coordinates $(q_{(\alpha)}^i, p_i^{(\alpha)})$, $i = 1, \cdots, n$. Then, we know that the dynamics consists of the orbits of a Hamiltonian vector field $X_H$. Every point of $M$ has an open neighborhood $U_\alpha$ with the local Darboux coordinates $(q_{(\alpha)}^i, p_i^{(\alpha)})$.

One can restrict the Hamiltonian $H$ and a nondegenerate two-form $\omega$ to each $U_\alpha$ to have a local Hamiltonian $H_\alpha = H_\alpha\big(q^i_{(\alpha)}, p^{(\alpha)}_i\big)$ and a symplectic structure $\omega_\alpha = dq^i_{(\alpha)} \wedge dp^{(\alpha)}_i$. Similarly the globally defined Hamiltonian vector field $X_H$ is restricted to $U_\alpha$ which is precisely given by $X_{H_\alpha}$. Then, the orbits are defined by Hamilton's equations

$$\frac{dq^i_{(\alpha)}}{dt} = \frac{\partial H_\alpha}{\partial p^{(\alpha)}_i}, \qquad \frac{dp^{(\alpha)}_i}{dt} = -\frac{\partial H_\alpha}{\partial q^i_{(\alpha)}}. \tag{A2}$$

When one takes the coordinate chart definition of symplectic manifolds, there is no compulsory reason to require the two-form $\omega$ to be closed. Indeed, the Hamiltonian formulation of particle dynamics consists of asking the local forms $\omega_\alpha$ and local functions $H_\alpha$ to glue on to a global symplectic form $\omega$ and a global Hamiltonian $H$. However, since the dynamical information is given by a global vector field, it is more natural to only require that the transition functions

$$q^i_{(\beta)} = q^i_{(\beta)}\big(q^i_{(\alpha)}, p^{(\alpha)}_i\big), \qquad p^{(\beta)}_i = p^{(\beta)}_i\big(q^i_{(\alpha)}, p^{(\alpha)}_i\big) \tag{A3}$$

on an overlap $U_\alpha \cap U_\beta \neq \varnothing$ preserve the form of Hamilton's equations (A2). This happens not only if Equation (A3) implies

$$\omega_\beta = dq^i_{(\beta)} \wedge dp^{(\beta)}_i = dq^i_{(\alpha)} \wedge dp^{(\alpha)}_i = \omega_\alpha, \qquad H_\beta = H_\alpha, \tag{A4}$$

where $H_\alpha : U_\alpha \to \mathbb{R}$, $\alpha \in I$, but also if it implies

$$\omega_\beta = \lambda_{\beta\alpha}\omega_\alpha, \qquad H_\beta = \lambda_{\beta\alpha}H_\alpha, \tag{A5}$$

where $\lambda_{\beta\alpha} = \text{constant} \neq 0$. Since $\iota(X_{H_\alpha})\omega_\alpha = dH_\alpha$, from Equation (A5) we obtain

$$X_{H_\alpha} = X_{H_\beta}, \tag{A6}$$

so the integral curves of $X_{H_\alpha}$ and $X_{H_\beta}$ are the same. Furthermore, Equation (A5) implies the cocycle condition

$$\lambda_{\gamma\beta}\lambda_{\beta\alpha} = \lambda_{\gamma\alpha} \tag{A7}$$

as the gluing condition. We know that the cocycle condition (A7) implies the existence of the local functions $\sigma_\alpha : U_\alpha \to \mathbb{R}$ satisfying

$$\lambda_{\beta\alpha} = \frac{e^{\sigma_\alpha}}{e^{\sigma_\beta}}. \tag{A8}$$

Thus, Equation (A5) shows that

$$\omega = e^{\sigma_\alpha}\omega_\alpha, \qquad H = e^{\sigma_\alpha}H_\alpha \tag{A9}$$

are globally defined on $M$. Moreover, a Hamiltonian vector field is globally defined, i.e., $X_H = X_{H_\alpha}$, as was indicated in Equation (A6). Hence, we have a basic line bundle $L$ over $M$ and a Hamiltonian $H$ as a cross-section of $L$ (a "twisted Hamiltonian") instead of a simple function. Therefore, $(M, \omega)$ is an LCS manifold that can be considered a natural phase space of Hamiltonian dynamical systems, more general than the symplectic manifolds.

As discussed in Section 2, the realization of emergent geometry is intrinsically local too. The emergent geometry is constructed by gluing local Darboux charts and their local Poisson algebras. Therefore, the construction of an LCS manifold as a generalized phase space for particle dynamics should also be applied to the emergent geometry. Therefore, we will briefly review infinitesimal automorphisms of an LCS manifold $(M, \Omega, b)$. The infinitesimal automorphism (IA) will be denoted by $\mathfrak{A}_\Omega$. Let $C^\infty(M)$ denote the associative

algebra of smooth functions on $M$ and $f : M \to \mathbb{R}$ be a globally defined function. The Hamiltonian vector field $X_f$ of $f \in C^\infty(M)$ with respect to the LCS form $\Omega$ is defined by

$$\iota(X_f)\Omega = df - fb. \tag{A10}$$

As we observed above, there is a well-defined line bundle $L$ over $M$ in which local functions $f_\alpha \equiv e^{-\sigma_\alpha} f$ on a patch $U_\alpha \subset M$ correspond to sections of $L \to U_\alpha$. If we take the Lee form on $U_\alpha$ as $b|_{U_\alpha} = d\sigma_\alpha$, Equation (A10) refers to the usual (local) Hamiltonian vector field $X_{f_\alpha} = X_f$ defined by

$$\iota(X_{f_\alpha})\Omega_\alpha = df_\alpha \tag{A11}$$

where $\Omega_\alpha = e^{-\sigma_\alpha}\Omega$. Using the Cartan formula for the Lie derivative

$$\mathcal{L}_X = d\iota_X + \iota_X d, \tag{A12}$$

one can immediately deduce from Equations (A1) and (A10) that

$$\mathcal{L}_{X_f}\Omega = b(X_f)\Omega, \tag{A13}$$
$$\mathcal{L}_{X_f}b = db(X_f). \tag{A14}$$

Therefore, unlike the symplectic case, the Hamiltonian vector field $X_f$ is, in general, not an IA of LCS manifolds.

Using the Hamiltonian vector fields defined by Equation (A10), we define the Poisson bracket as

$$\{f, g\}_\Omega = \iota(X_f)\iota(X_g)\Omega = -\Omega(X_f, X_g) = e^{\sigma_\alpha}\iota(X_{f_\alpha})\iota(X_{g_\alpha})\Omega_\alpha = e^{\sigma_\alpha}\{f_\alpha, g_\alpha\}_{\Omega_\alpha}. \tag{A15}$$

Then, we can calculate the double Poisson bracket

$$\{\{f, g\}_\Omega, h\}_\Omega = X_h\big(\Omega(X_f, X_g)\big) - b(X_h)\Omega(X_f, X_g). \tag{A16}$$

Using this result, it is easy to check the Jacobi identity of the Poisson bracket:

$$\{\{f, g\}_\Omega, h\}_\Omega + \{\{g, h\}_\Omega, f\}_\Omega + \{\{h, f\}_\Omega, g\}_\Omega = \big(d\Omega - b \wedge \Omega\big)(X_f, X_g, X_h) = 0. \tag{A17}$$

Let $\mathfrak{P} = (C^\infty(M), \{-, -\}_\Omega)$ be the Poisson–Lie algebra of $(M, \Omega)$ and $\mathfrak{X}(M)$ the Lie algebra of vector fields on $M$. The result (A15) shows that the mapping $\mathfrak{H} : \mathfrak{P} \to \mathfrak{X}(M)$ given by $f \mapsto X_f$ is a Lie algebra homomorphism because one can derive the relation

$$X_{\{f, g\}_\Omega} = [X_f, X_g] \tag{A18}$$

from the Jacobi identity (A17). However, if $(M, \Omega)$ is a (connected) LCS manifold that is not GCS, then $\mathfrak{H}$ must be a monomorphism, i.e., an injective homomorphism. See Proposition 2.1 in [74] for the proof. This means that $X_f = 0$ implies $f = 0$. This is in stark contrast to symplectic manifolds, in which $X_f = 0$ just implies $f = $ constant. Since we claim that the phase space for cosmic inflation is a locally conformal (co)symplectic manifold, this reveals a remarkable property that vacuum energy couples to gravity and triggers cosmic inflation, as we noted before. However, it does not mean that the cosmological constant couples to gravity because physical quantities during inflation are not constant but time-dependent.

Denote the IA of $(M, \Omega)$ by $\mathfrak{X}_\Omega(M)$ whose elements obey $\mathcal{L}_X\Omega = 0$. Then, we have $\mathcal{L}_X b = 0$ by Equation (A1) which implies the condition $b(X) = $ constant. In particular, if $X, Y \in \mathfrak{X}_\Omega(M)$, then $b(X) = $ constant, $b(Y) = $ constant and $db(X, Y) = 0$ yields $b([X, Y]) = 0$ using the formula

$$db(X, Y) = X\big(b(Y)\big) - Y\big(b(X)\big) - b([X, Y]). \tag{A19}$$

Hence, the application $l : \mathfrak{X}_\Omega(M) \to \mathbb{R}$ defined by $l(X) = b(X)$ is a Lie algebra homomorphism, called the Lee homomorphism of $\mathfrak{X}_\Omega(M)$. The kernel $\ker(l)$ is the Lie algebra of the horizontal elements of $\mathfrak{X}_\Omega(M)$, denoted by $\mathfrak{X}_\Omega^{\text{hor}}(M)$. The IA $X \in \mathfrak{X}_\Omega(M)$ with $l(X) \neq 0$ is called transversal IA and an LCS manifold $M$ is called the first kind if it has a transversal IA. Otherwise, $M$ is of the second kind and the Lee homomorphism is trivial. Note that, if $(M, \Omega)$ is of the first kind and $f : M \to \mathbb{R}$ is a function such that $df|_{x_0} = b(x_0)$, then $(M, e^{-f}\Omega)$ has the Lee form $b - df$ with a vanishing point, so it becomes an LCS manifold of the second kind.

There is a special vector field $A$ defined by $\iota_A \Omega = b$. Then, it is easy to see

$$\iota_A b = 0, \qquad \mathcal{L}_A b = 0, \qquad \mathcal{L}_A \Omega = 0. \tag{A20}$$

We do have $X_f \in \mathfrak{X}_\Omega(M)$ if and only if $b(X_f) = 0$ according to Equation (A13) or, equivalently, $b(X_f) = \iota_{X_f} \iota_A \Omega = -\iota_A(df - fb) = -A(f) = 0$. Let us fix an element $B \in l^{-1}(1) \subset \mathfrak{X}_\Omega(M)$. Then, every element $Y$ in $\mathfrak{X}_\Omega(M)$ has a unique decomposition

$$Y = X + l(Y)B, \qquad X \in \mathfrak{X}_\Omega^{\text{hor}}(M). \tag{A21}$$

Now, put $a \equiv -\iota_B \Omega$, so $a(B) = 0$ and $a(A) = \iota_B \iota_A \Omega = b(B) = 1$. Since $\mathcal{L}_B \Omega = (\iota_B d + d\iota_B)\Omega = 0$, this yields a particular expression for $\Omega$ given by

$$\Omega = da - b \wedge a = d_b a, \tag{A22}$$

where $d_b$ is the Lichnerowicz differential defined by $d_b \beta = d\beta - b \wedge \beta$ for any $k$-form $\beta$ and satisfies $d_b^2 = 0$. Furthermore, using the formula $[\mathcal{L}_X, \iota_Y] = \iota_{[X,Y]}$ for vector fields $X$ and $Y$, we have $\mathcal{L}_B a = 0$, and hence, $\iota_B da = 0$, which means rank $da < 2n$. Since $\Omega^n \neq 0$, one can deduce from Equation (A22) the condition

$$b \wedge a \wedge (da)^{n-1} \neq 0 \tag{A23}$$

everywhere. This yields Proposition 2.2 in Ref. [74] that a manifold $M$ of dimension $2n$ admits an LCS structure of the first kind if and only if it admits two one-forms $a, b$ such that $db = 0$, rank $da < 2n$ and Equation (A23) holds at every point of $M$. Note also that $\iota_A da = \iota_A(\Omega + b \wedge a) = b - a(A)b = 0$. This means that $[A, B] = 0$ because $\iota_A da = \mathcal{L}_A a = -\mathcal{L}_A \iota_B \Omega = -\iota_{[A,B]}\Omega = 0$. In sum, there exist particular vector fields $A$ and $B$ in $\mathfrak{X}_\Omega(M)$ that obey

$$[A, B] = 0, \qquad a(A) = b(B) = 1, \qquad a(B) = b(A) = 0. \tag{A24}$$

Thus, one can obtain on $M$ the vertical foliation $\mathcal{V} = \text{span}\{A, B\}$, whose leaves are the orbits of a natural action of $\mathbb{R}^2$.

Suppose that $(M, \Omega)$ is an LCS manifold of the first kind and $B$ is a basic transversal IA. Let $\mathfrak{X}_\Omega^{\text{hor}}(M, B)$ be the Lie subalgebra of $\mathfrak{X}_\Omega^{\text{hor}}(M)$ whose automorphisms also preserve $B$. It turns out that $X \in \mathfrak{X}_\Omega^{\text{hor}}(M, B)$ if and only if $\mathcal{L}_X \Omega = 0$, $b(X) = 0$ and $[X, B] = 0$. Similarly, consider the subset of $C^\infty(M)$ that consists of functions satisfying $A(f) = B(f) = 0$ and is denoted by $C_\mathcal{V}^\infty(M)$. Then, one can show that $\mathfrak{P}_\mathcal{V} = (C_\mathcal{V}^\infty(M), \{-, -\}_\Omega)$ is a Poisson–Lie subalgebra of $\mathfrak{P}$ and $\mathfrak{H} : \mathfrak{P}_\mathcal{V} \to \mathfrak{X}_\Omega^{\text{hor}}(M, B)$ is an isomorphism. A striking fact is that a semi-simple Lie group $G$ cannot act transitively on a nonsymplectic LCS manifold.

Formula (A13) proves that a Hamiltonian vector field is a conformal infinitesimal transformation (CIT) of $(M, \Omega)$. In general, a vector field $X$ is a CIT if

$$\mathcal{L}_X \Omega = \alpha_X \Omega \tag{A25}$$

where $\alpha_X$ is a function on $M$. The CIT forms a Lie algebra denoted by $\mathfrak{X}_\Omega^c(M)$. By differentiating Equation (A25), one can derive that $\mathcal{L}_X b = d\alpha_X$, which implies

$$\alpha_X = b(X) + \kappa, \qquad \kappa = \text{constant}. \tag{A26}$$

One can rewrite Equation (A25) as

$$\kappa \Omega = d_b(\iota_X \Omega). \tag{A27}$$

Thus, an LCS form $\Omega$ is $d_b$-exact if there is a CIT $X$, or it can be written in terms of a local symplectic form $\Omega_\alpha = e^{-\sigma_\alpha} \Omega$ as

$$\mathcal{L}_X \Omega_\alpha = (\alpha_X - b(X))\Omega_\alpha. \tag{A28}$$

That is, the local form of the CIT is given by

$$\mathcal{L}_X \Omega_\alpha = \kappa \Omega_\alpha. \tag{A29}$$

If we write $\Omega_\alpha = dA_{(\alpha)}$ on an open neighborhood $U_\alpha$ according to the Poincaré lemma, Equation (A29) can be written as the form [73]

$$\iota_X \Omega_\alpha = \kappa A_{(\alpha)} + df_\alpha, \tag{A30}$$

where $f_\alpha : U_\alpha \to \mathbb{R}$ is a smooth function on $U_\alpha$. If the conditions (A29) and (A30) hold either locally or globally, we will call $X$ a conformal vector field which plays an important role in our discussion. If $H^1(M) = 0$, the conformal vector field $X$ has a unique decomposition given by

$$X = \kappa Z + X_f, \tag{A31}$$

where $\iota_Z \Omega = e^{\sigma_\alpha} A_{(\alpha)} \equiv A$ and $\iota_{X_f} \Omega = df - fb$. The vector field $Z$ is called the Liouville vector field [72]. Note that, even though $f = 0$ identically, the conformal vector field $X = \kappa Z$ is nontrivial and it is generated by the open Wilson line (112) in our case. We observed in Section 3 that this property leads to a remarkable consequence for the cosmic inflation.

We can extend the Lee homomorphism to $l : \mathfrak{X}_\Omega^c(M) \to \mathbb{R}$ by defining $l(X) = b(X) - \alpha_X = -\kappa$. If $X, Y \in \mathfrak{X}_\Omega^c(M)$, we obtain $l([X,Y]) = b([X,Y]) - \alpha_{[X,Y]} = -\kappa$ from $\mathcal{L}_{[X,Y]}\Omega = \alpha_{[X,Y]}\Omega$. Hence, the extended $l$ is also a Lie algebra homomorphism. Its kernel is denoted by $\ker l = \mathfrak{X}_{\text{Ham}}^l(M)$ and consists of vector fields $X$ to obey $\mathcal{L}_X \Omega_\alpha = 0$, i.e., of locally Hamiltonian vector fields. Note that $\tilde{l}(X)$ for $\tilde{\Omega} = e^\varphi \Omega$ is equal to $l(X)$ for $\Omega$. Thus, the Lee homomorphism $l$ is conformally invariant. If we fix an element $C \in l^{-1}(1)$, we can obtain for every $Y \in \mathfrak{X}_\Omega^c(M)$ the unique decomposition

$$Y = X + l(Y)C, \qquad X \in \mathfrak{X}_{\text{Ham}}^l(M). \tag{A32}$$

Then, if $c = -\iota_C \Omega$, we can solve $\mathcal{L}_C \Omega = (\iota_C d + d\iota_C)\Omega = \alpha_C \Omega$ to obtain a particular expression for $\Omega$ given by

$$\Omega = dc - b \wedge c = d_b c. \tag{A33}$$

In a conservative dynamical system described by a Hamiltonian vector field, time coordinate $t$ is not a phase space coordinate but an affine parameter on particle trajectories. However, for a general time-dependent system, it is useful to include the time coordinate as an extra phase space coordinate. The corresponding $(2n + 1)$-dimensional manifold is known as an almost cosymplectic manifold which is a triple $(M, \Omega, \eta)$ where $\Omega$ and $\eta$ are a two-form and a one-form on $M$ such that $\eta \wedge \Omega^n \neq 0$. If $\Omega$ and $\eta$ are closed, i.e., $d\Omega = d\eta = 0$, then $M$ is said to be a cosymplectic manifold. Thus, an odd-dimensional counterpart of a symplectic manifold is given by a cosymplectic manifold, which is locally a product of a symplectic manifold with a circle or a line. A contact manifold constitutes a subclass of cosymplectic manifolds with $\Omega = d\eta$ [60,61,68]. Then, the one-form $\eta$ is called a

contact structure or a contact one-form. Given a contact one-form $\eta$, there is a unique vector field $R$ such that $\iota_R \eta = 1$ and $\iota_R \Omega = 0$. This vector field $R$ is known as the Reeb vector field of the contact form $\eta$. Two contact forms $\eta$ and $\eta'$ on $M$ are equivalent if there is a smooth positive function $\rho$ on $M$ such that $\eta' = \rho\eta$, since $\eta' \wedge (d\eta')^n = \rho^{n+1} \eta \wedge (d\eta)^n \neq 0$. The contact structure $C(\eta)$ determined by $\eta$ is the equivalence class of $\eta$.

The Darboux theorem for a contact manifold $(M, \eta)$ states [60,61,68] that, in an open neighborhood of each point of $M$, it is always possible to find a set of local (Darboux) coordinates $(x^1, \cdots, x^n, y_1, \cdots, y_n, z)$ such that the one-form $\eta$ can be written as

$$\eta = dz - \sum_{i=1}^{n} y_i dx^i \tag{A34}$$

and the Reeb vector field is given by

$$R = \frac{\partial}{\partial z}. \tag{A35}$$

To understand the contact one-form $\eta$ more closely, first let us denote by $\mathcal{D}$ the contact distribution or subbundle defined by the kernel of $\eta$. If $X, Y$ are (local) vector fields in $\mathcal{D}$, we have

$$d\eta(X, Y) = X\big(\eta(Y)\big) - Y\big(\eta(X)\big) - \eta([X, Y]) = -\eta([X, Y]). \tag{A36}$$

This says that the distribution is integrable if and only if $d\eta$ is zero on $\mathcal{D}$. However, the condition $\eta \wedge (d\eta)^n \neq 0$ means that the kernel of $d\eta$ is one-dimensional and everywhere transverse to $\mathcal{D}$. Consequently, $d\eta$ is a linear symplectic form on $\mathcal{D}$ and the largest integral submanifolds of $\mathcal{D}$ are $n$-dimensional, so maximally non-integrable. In other words, a contact structure is nowhere integrable. In the above Darboux coordinate system, the contact subbundle $\mathcal{D}$ is spanned by

$$X_i = \frac{\partial}{\partial x^i} + y_i \frac{\partial}{\partial z}, \qquad Y^i = \frac{\partial}{\partial y_i}, \qquad i = 1, \cdots, n, \tag{A37}$$

so they obey the bracket relations

$$[X_i, Y^j] = -\delta_i^j R, \qquad [X_i, R] = [Y^i, R] = 0. \tag{A38}$$

Since $d\eta = \sum_{i=1}^{n} dx^i \wedge dy_i$ is a symplectic form with rank $2n$, the kernel of $d\eta$ is one-dimensional and generated by the Reeb vector $R$. Therefore, every vector field $X$ on $M$ can be uniquely written as $X = fR + Y$ where $f \in C^\infty(M)$ and $Y$ is a section of $\mathcal{D}$. A contact structure is regular if $R$ is regular as a vector field, that is, every point of the manifold has a neighborhood such that any integral curve of the vector field passing through the neighborhood passes through only once.

Given a $(2n - 1)$-dimensional contact manifold $M$ with a contact form $a$, i.e., $a \wedge (da)^{n-1} \neq 0$, one can construct an LCS manifold by considering a principal bundle $p : V \to M$ with group $\mathbb{S}^1$ over $M$. Consider $V = \mathbb{S}^1 \times M$ endowed with the form $\Omega = da - b \wedge a = d_b a$, where $b$ is the canonical one-form on $\mathbb{S}^1$. Clearly, $\Omega$ is nondegenerate and $b$ is closed but not exact, and it obeys $d\Omega - b \wedge \Omega = d_b \Omega = d_b^2 a = 0$. Hence, $(V, \Omega)$ is an LCS manifold having $b$ as its Lee form, but it is not GCS. More generally, let $p : V \to M$ be an arbitrary principal bundle with group $\mathbb{S}^1$ over a $(2n - 1)$-dimensional manifold $M$, and let $a$ be the connection one-form on this principal bundle and $F = da$ be the corresponding curvature two-form. Then, if $b \wedge a \wedge F^{n-1} \neq 0$, the form $\Omega = F - b \wedge a$ defines an LCS structure on $V$ which is not GCS.

Let $\mathfrak{X}(M)$ and $\Lambda^1(M)$ be the $C^\infty(M)$-modules of differentiable vector fields and one-forms on $M$, respectively. If $(M, \Omega, \eta)$ is a cosymplectic manifold, then there exists an isomorphism of $C^\infty(M)$-modules

$$\Upsilon : \mathfrak{X}(M) \to \Lambda^1(M) \tag{A39}$$

defined by

$$Y(X) = \iota_X \Omega + \eta(X)\eta. \tag{A40}$$

The Reeb vector field is given by $R = Y^{-1}(\eta)$. Let $f : M \to \mathbb{R}$ be a smooth function on $M$. The Hamiltonian vector field $X_f$ is then defined by

$$Y(X_f) = df - R(f)\eta + \eta. \tag{A41}$$

In other words, $X_f$ is the vector field characterized by the identities

$$\iota(X_f)\Omega = df - R(f)\eta, \qquad \eta(X_f) = 1. \tag{A42}$$

Then, one can check that the time-like vector field $V_0$ in Equation (67) is a Hamiltonian vector field for a cosymplectic manifold $(\mathbb{R} \times M, \pi_2^* B, dt)$ where $\pi_2 : \mathbb{R} \times M \to M$ and $(M, B)$ is a symplectic manifold.

An almost cosymplectic manifold $(M, \Omega, \eta)$ is said to be LCC if there exist an open covering $\{U_\alpha\}_{\alpha \in I}$ and local functions $\sigma_\alpha : U_\alpha \to \mathbb{R}$ such that

$$d(e^{-\sigma_\alpha}\Omega) = 0, \qquad d(e^{-\sigma_\alpha}\eta) = 0. \tag{A43}$$

The local one-forms $d\sigma_\alpha$ glue to a closed one-form $b$ satisfying

$$d\Omega - b \wedge \Omega = d_b\Omega = 0, \qquad d\eta - b \wedge \eta = d_b\eta = 0. \tag{A44}$$

Two LCC structures $(\Omega', \eta')$ and $(\Omega, \eta)$ are equivalent if $\Omega' = f\Omega$ and $\eta' = f\eta$ for a positive function $f$ on $M$ where the Lee form of $\Omega'$ is given by $b' = b + d \ln f$. An LCC manifold reduces to a cosymplectic manifold if the Lee form $b$ vanishes, while it becomes an LCS manifold if $\eta = 0$ identically. The isomorphism (A40) can be generalized to LCC manifolds and the corresponding Hamiltonian vector field is defined by

$$X_f = Y^{-1}(df - R(f)\eta + \eta) + fS \tag{A45}$$

where $S$ is called the canonical vector field defined by

$$Y(S) = b(R)\eta - b. \tag{A46}$$

Therefore, $X_f$ is characterized by the identities

$$\iota(X_f)\Omega = df - R(f)\eta + f(b(R)\eta - b), \qquad \eta(X_f) = 1. \tag{A47}$$

It was shown in [75] that an LCC manifold can be seen as a generalized phase space of time-dependent Hamiltonian systems. Hence, we argue that an LCC manifold also corresponds to a generalized phase space for an inflationary universe and its quantization realizes a background-independent formulation of the cosmic inflation, in particular, in the context of emergent spacetime.

## Appendix B. Harmonic Oscillator with Time-Dependent Mass

We observed that the NC spacetime $\mathbb{R}_\theta^{2n}$ in equilibrium is described by the Hilbert space of an $n$-dimensional harmonic oscillator, while the inflating spacetime in nonequilibrium is described by the $n$-dimensional harmonic oscillator with negative friction. The corresponding harmonic oscillator of constant frequency $\omega$ and friction coefficient $\alpha$ satisfies the equation

$$\ddot{q}^i + 2\alpha\dot{q}^i + \omega^2 q^i = 0, \qquad i = 1, \cdots, n. \tag{A48}$$

The inflationary coordinates (99) correspond to the case $\alpha = -\frac{\kappa}{2} < 0$. It is known that the above second-order equation of motion cannot be directly derived from the Euler–Lagrange equation of any Lagrangian. However, there is an equivalent second-order equation

$$e^{2\alpha t}(\ddot{q}^i + 2\alpha \dot{q}^i + \omega^2 q^i) = 0, \tag{A49}$$

for which a variational principle can be found [93]. Although Equation (A48) is traditionally considered to be non-Lagrangian, there exists an action principle for the equation of motion (A49) in terms of the Lagrangian

$$L = \frac{1}{2}m(\dot{q}^2 - \omega^2 q^2)e^{2\alpha t}. \tag{A50}$$

The corresponding Hamiltonian is given by

$$H = \frac{1}{2m}(e^{-2\alpha t}p^2 + e^{2\alpha t}m^2\omega^2 q^2) \tag{A51}$$

where $p_i = m\dot{q}^i e^{2\alpha t}$.

It is interesting to note that the equation of motion (A49) can be derived from an $n$-dimensional harmonic oscillator with a time-dependent mass $m(t)$ whose action is given by

$$S = \frac{1}{2}\int dt \left(m(t)\dot{q}^2 - k(t)q^2\right) \tag{A52}$$

where $k(t) = m(t)\omega^2$ with constant frequency $\omega$. The variational principle, $\delta S = 0$, with respect to arbitrary variations $\delta q^i$ leads to the equation of motion

$$m(t)\left(\ddot{q}^i + \frac{\dot{m}(t)}{m(t)}\dot{q}^i + \omega^2 q^i\right) = 0. \tag{A53}$$

The second-order Equation (A49) corresponds to the case

$$\frac{\dot{m}(t)}{m(t)} = 2\alpha \quad \Rightarrow \quad m(t) = m_0 e^{2\alpha t}. \tag{A54}$$

Recall that the equation of motion for the inflaton field corresponds to the case with the time-dependent mass $m(t) = m_0 e^{3Ht}$.

There is also the first-order formalism for the dynamical system (A52). The action has the form

$$S = \frac{1}{2}\int dt \left(y\dot{x} - x\dot{y} - (y^2 + 2\alpha xy + \omega^2 x^2)\right)e^{2\alpha t}. \tag{A55}$$

The equations of motion derived from the action (A55) are given by

$$(\dot{y} + 2\alpha y + \omega^2 x)e^{2\alpha t} = 0, \qquad (\dot{x} - y)e^{2\alpha t} = 0. \tag{A56}$$

The above action (A55) describes a singular system with second-class constraints

$$\phi_x = p_x - \frac{1}{2}ye^{2\alpha t}, \qquad \phi_y = p_y + \frac{1}{2}xe^{2\alpha t} \tag{A57}$$

with the Hamiltonian

$$H(x, y, t) = \frac{1}{2}(y^2 + 2\alpha xy + \omega^2 x^2)e^{2\alpha t}. \tag{A58}$$

Even though the constraints are explicitly time-dependent, it is still possible to apply the Hamiltonian formalism with the help of Dirac brackets and perform the canonical quantization of the system. It was shown in [93] that the classical and quantum description of the harmonic oscillator described by the action (A52) is equivalent to the first-order approach

given in terms of the constraint system described by the action (A55). Furthermore, it can be proved that the dynamical system described by Equation (A49) is locally (i.e., $|t| < \infty$) equivalent to the system with the equation of motion (A48).

**Appendix C. NC Spacetime as a Second-Quantized String**

We know that quantum mechanics is a more fundamental description of nature than classical physics. The microscopic world is already quantum. Nevertheless, the quantization is necessary to find a quantum theoretical description of nature since we have understood our world starting with the classical description which we understand better. After quantization, the quantum theory is described by a fundamental NC algebra such as Equation (34). A striking feature of the NC algebra $\mathcal{A}_\hbar$ is that every point in $\mathbb{R}^n$ is unitarily equivalent because translations in $\mathbb{R}^n$ are generated by an inner automorphism of $\mathcal{A}_\hbar$, i.e., $f(x + a) = U(a)f(x)U(a)^\dagger$ where $f(x) \in \mathcal{A}_\hbar$ and $U(a) = e^{ip_i a^i/\hbar} \in \mathrm{Inn}(\mathcal{A}_\hbar)$. Therefore, through the quantization, the concept of (phase) space is doomed. Instead the (phase) space is replaced by the algebra $\mathcal{A}_\hbar$ and its Hilbert space representation and dynamical variables become operators acting on the Hilbert space. Only in the classical limit, is a phase space with the symplectic structure $\omega = dx^i \wedge dp_i$ emergent from the quantum algebra $\mathcal{A}_\hbar$ such as (34).

Recall that the mathematical structure of NC spacetime is basically the same as the NC phase space in quantum mechanics [94]. Therefore, essential features in quantum mechanics must be applied to the NC spacetime too. In particular, NC algebras $\mathcal{A}_\theta$ such as the NC space (19) also play a fundamental role and every point in the NC space is indistinguishable, i.e., unitarily equivalent because any two points are connected by an inner automorphism of $\mathcal{A}_\theta$. This implies that there is no concept of space(time) in the NC algebra $\mathcal{A}_\theta$ for the same reason as quantum mechanics and a classical spacetime must be derived from the NC algebra $\mathcal{A}_\theta$. After all, an important lesson is that NC spacetime necessarily implies emergent spacetime.

Although spacetime at a microscopic scale, e.g., the Planck scale $L_P$, is intrinsically NC, we understand the NC spacetime through the quantization of a symplectic (or, more generally, Poisson) manifold. Let $(M, B)$ be a symplectic manifold. On the one hand, the basic concept in symplectic geometry is an area defined by the symplectic two-form $B$ that is a nondegenerate, closed two-form. On the other hand, the basic concept in Riemannian geometry determined by a pair $(M, g)$ is a distance defined by the metric tensor $g$ that is a nondegenerate, symmetric bilinear form. One may identify this distance with a geodesic worldline of a "particle" moving in $M$. Geodesic curves in $M$ gives us all the information about Riemannian geometry $(M, g)$. Conversely, the area in symplectic geometry $(M, B)$ may be regarded as a minimal worldsheet swept by a "string" moving in $M$. In this picture, the wiggly string, so a fluctuating worldsheet, corresponds to a deformation of symplectic structure in $M$. This picture becomes more transparent by the so-called pseudoholomorphic or $J$-holomorphic curve introduced by Gromov [95].

Let $(M, J)$ be an almost complex manifold and $(\Sigma, j)$ be a Riemann surface. By the compatibility of $J$ to $B$, we have the relation $g(X, Y) = B(X, JY)$ for any vector fields $X, Y \in \mathfrak{X}(M)$. Let us also fix a Hermitian metric $h$ of $(\Sigma, j)$. A smooth map $f : \Sigma \to M$ is called pseudoholomorphic [50] if the differential $df : T\Sigma \to TM$ is a complex linear map with respect to $j$ and $J$:

$$df \circ j = J \circ df. \tag{A59}$$

This condition corresponds to the commutativity of the following diagram

$$
\begin{array}{ccc}
T\Sigma & \xrightarrow{\ j\ } & T\Sigma \\
{\scriptstyle df}\downarrow & & \downarrow{\scriptstyle df} \\
TM & \xrightarrow[\ J\ ]{} & TM
\end{array}
$$

Since $J^{-1} = -J$, it is also equivalent to $\bar{\partial}_J f = 0$ where $\bar{\partial}_J f := \frac{1}{2}(df + J \circ df \circ j)$. For example, suppose that the Riemann surface is $(\Sigma, i)$ where $i$ is the standard complex structure. We can work in a chart $u_\epsilon : U_\epsilon \to \mathbb{C}$ with local coordinate $z = \tau + i\sigma$ where $U_\epsilon \subset \Sigma$ is an open neighborhood. Define $f_\epsilon = f \circ u_\epsilon^{-1}$. In this case, we have

$$\bar{\partial}_J f = \frac{1}{2}\left[\left(\frac{\partial f_\epsilon}{\partial \tau} + J(f_\epsilon)\frac{\partial f_\epsilon}{\partial \sigma}\right)d\tau + \left(\frac{\partial f_\epsilon}{\partial \sigma} - J(f_\epsilon)\frac{\partial f_\epsilon}{\partial \tau}\right)d\sigma\right]. \tag{A60}$$

Thus, we see that $\bar{\partial}_J f = 0$ if

$$\frac{\partial f_\epsilon}{\partial \tau} + J(f_\epsilon)\frac{\partial f_\epsilon}{\partial \sigma} = 0. \tag{A61}$$

Since $J$ is $B$-compatible, every smooth map $f : \Sigma \to M$ satisfies [96–98]

$$\frac{1}{2}\int_\Sigma ||df||_g^2 \, d\text{vol}_\Sigma = \int_\Sigma ||\bar{\partial}_J f||_g^2 \, d\text{vol}_\Sigma + \int_\Sigma f^* B, \tag{A62}$$

where the norms are taken with respect to the metric $g$ and $d\text{vol}_\Sigma$ is a volume form on $\Sigma$. In terms of local coordinates, $(\sigma^1, \sigma^2)$ on $\Sigma$ and $f(\sigma) = (x^1, \cdots, x^{2n})$ on $M$,

$$||df||_g^2 = g_{\mu\nu}(f(\sigma))\frac{\partial x^\mu}{\partial \sigma^\alpha}\frac{\partial x^\nu}{\partial \sigma^\beta}h^{\alpha\beta}(\sigma) \tag{A63}$$

and $d\text{vol}_\Sigma = \sqrt{h}d^2\sigma$. Therefore, the left-hand side of Equation (A62) is nothing but the Polyakov action in string theory. For a pseudoholomorphic curve $f : \Sigma \to M$ that obeys $\bar{\partial}_J f = 0$, we thus have the identity

$$S_P(f) \equiv \frac{1}{2}\int_\Sigma ||df||_g^2 \, d\text{vol}_\Sigma = \int_\Sigma f^* B. \tag{A64}$$

This means that any pseudoholomorphic curves minimize the "harmonic energy" $S_P(f)$ in a fixed homology class and so are harmonic maps. In other words, their symplectic area coincides with the surface area. Therefore, any pseudoholomorphic curve is a solution of the worldsheet Polyakov action $S_P(f)$. For instance, if $M = \mathbb{C}^n$ with complex coordinates $\phi^i = x^{2i-1} + \sqrt{-1}x^{2i}$ $(i = 1, \ldots, n)$ and $f_\epsilon(z, \bar{z}) \equiv \phi^i(z, \bar{z})$, Equation (A61) becomes

$$\frac{1}{2}\left(\frac{\partial}{\partial \tau} + \sqrt{-1}\frac{\partial}{\partial \sigma}\right)\phi^i(z, \bar{z}) = \partial_{\bar{z}}\phi^i(z, \bar{z}) = 0. \tag{A65}$$

In this case, pseudoholomorphic curves coincide with holomorphic curves. Moreover, such curves are harmonic and minimal surfaces.[16]

The pseudoholomorphic curve also provides us with a useful tool to understand the emergent gravity picture. To demonstrate this aspect, let us include a boundary interaction in the sigma model (A62) such that the open string action is given by

$$S_A(f) \equiv \frac{1}{2}\int_\Sigma ||df||_g^2 \, d\text{vol}_\Sigma + \int_{\partial\Sigma} f^* A, \tag{A66}$$

where the one-form $A$ is the connection of a line bundle $L \to M$. Using Stokes' theorem, the second term can be written as

$$\int_{\partial\Sigma} f^* A = \int_\Sigma f^* dA. \tag{A67}$$

After combining the identities (A62) and (A67) together, we write the action

$$S_A(f) = \int_\Sigma ||\bar{\partial}_J f||_g^2 \, d\text{vol}_\Sigma + \int_\Sigma f^* \mathcal{F}, \tag{A68}$$

where $\mathcal{F} = B + F$ and $F = dA$. If one recalls the derivation of Equation (A62), one may immediately realize that the action $S_A(f)$ can equivalently be written in the form of the Polyakov action

$$S_P(\psi) \equiv \frac{1}{2} \int_\Sigma ||d\psi||_{\mathcal{G}}^2 \, d\text{vol}_\Sigma, \tag{A69}$$

where the differential $d\psi$ for a smooth map $\psi : \Sigma \to M$ has the norm taken with respect to some metric $\mathcal{G}$. For this purpose, let us assume that the almost complex structure $J$ is also compatible with the deformed symplectic structure $\mathcal{F}$, i.e.,

$$\mathcal{G}(X, Y) = \mathcal{F}(X, JY), \qquad \forall X, Y \in \mathfrak{X}(M) \tag{A70}$$

is a Riemannian metric on $M$. An explicit representation of the Polyakov action (A69) can be made by introducing local coordinates $\psi(\sigma) = (X^1, \cdots, X^{2n})$ on an open set $U_i \subset M$ so that

$$||d\psi||_{\mathcal{G}}^2 = \mathcal{G}_{\mu\nu}(\psi(\sigma)) \frac{\partial X^\mu}{\partial \sigma^\alpha} \frac{\partial X^\nu}{\partial \sigma^\beta} h^{\alpha\beta}(\sigma). \tag{A71}$$

One can then apply the same derivation of Equation (A62) to the action (A69) to derive the identity

$$\frac{1}{2} \int_\Sigma ||d\psi||_{\mathcal{G}}^2 \, d\text{vol}_\Sigma = \int_\Sigma ||\bar{\partial}_J \psi||_{\mathcal{G}}^2 \, d\text{vol}_\Sigma + \int_\Sigma \psi^* \mathcal{F}. \tag{A72}$$

For pseudoholomorphic curves $\psi : \Sigma \to M$ satisfying $\bar{\partial}_J \psi = 0$, we finally obtain the result

$$S_P(\psi) = \frac{1}{2} \int_\Sigma ||d\psi||_{\mathcal{G}}^2 \, d\text{vol}_\Sigma = \int_\Sigma \psi^* \mathcal{F}. \tag{A73}$$

The above argument reveals a nice picture in that dynamical $U(1)$ gauge fields in a line bundle $L$ over $M$ deform an underlying symplectic structure $(M, B)$ and this deformation is transformed into the dynamics of gravity [27]. As we observed before, the symplectic geometry is probed by strings, while the Riemannian geometry is probed by particles. We note that the NC space (19) defines only a minimal area $\alpha' = l_s^2$, whereas the concept of point is doomed if $\hbar$ in quantum mechanics introduces a minimal area in the NC phase space (34). The minimal area (surface) in the NC space behaves like the smallest unit of spacetime blob and acts as a basic building block of string theory. The concept of pseudoholomorphic or $J$-holomorphic curves in symplectic geometry plays a role in such minimal surfaces. It is known [50] that there is a nonlinear Fredholm theory which describes the deformations of a given pseudoholomorphic curve $f : \Sigma \to (M, J)$ and the deformations are parameterized by a finite-dimensional moduli space. (This moduli space may be enriched by considering pseudoholomorphic curves in an LCS manifold.) When a symplectic manifold is probed with a string or pseudoholomorphic curve, the notion of a wiggly string in this probe picture corresponds to the deformation of a symplectic structure. Hence, the emergence of gravity from symplectic geometry or more precisely NC $U(1)$ gauge fields may be reasonable because we know from string theory that a Riemannian geometry (or general relativity) is emergent from the wiggly string.

We can think of the integral $A(f) = \int_\Sigma f^* B$ in two ways if $f$ is a pseudoholomorphic curve. On the one hand, the pointwise compatibility between the structures $(B, J)$ means that $A(f)$ is essentially the area of the image of $f$, measured in the Riemannian metric $g$. On the other hand, the condition that $B$ is closed means that $A(f)$ is a topological (homotopy) invariant of the map $f$ since it depends only on the evaluation of a closed 2-form $B$ on the 2-chain defined by $f(\Sigma)$. Hence, we can use the curves in two main ways [50]. The first way is as geometrical probes to explore a symplectic manifold, as we advocated above. The second way is as the source of numerical invariants known as the Gromov–Witten invariants. Using the pseudoholomorphic curves, Gromov proved a surprising non-squeezing theorem [95–98] stating that a ball $B_{2n}(r)$ of radius $r$ in a symplectic vector space $\mathbb{R}^{2n}$ with the standard symplectic form $B$ cannot be mapped by a symplectomorphism

into any cylinder $B_2(R) \times \mathbb{R}^{2n-2}$ of radius $R$ if $R < r$. It is possible to replace $\mathbb{R}^{2n-2}$ by a $(2n-2)$-dimensional compact symplectic manifold $V$ with $\pi_2(V) = 0$.

Now, we will discuss how an NC space provides us an important clue for a background-independent formulation of string theory. The NC spacetime is defined by the quantization of a symplectic manifold $(M, B)$. One may try to lift the notion of the pseudoholomorphic curve to a quantized symplectic manifold, namely, an NC space such as Equation (19). The quantization of a symplectic manifold leads to a radical change in classical concepts such as spaces and observables. The classical space is replaced by a Hilbert space and dynamical observables become operators acting on the Hilbert space. Then, as we discussed in Section 2, the NC spacetime will provide a more elegant framework for the background-independent formulation of quantum gravity in terms of matrix models, which is still elusive in string theory. Recall that the dynamical Lorentzian spacetime (64) emerges from a classical solution of the matrix model (68), and the cosmic inflation described by the metric (124) also arises as a solution of the time-dependent matrix model.

In order to grasp how a pseudoholomorphic curve looks like in NC spacetime, let us consider the simplest case in Equation (A65). After quantization, the coordinates of $\mathbb{C}^n$ denoted by $\phi^i(z, \bar{z})$ become operators in an NC $\star$-algebra $\mathcal{A}_\theta^2 \equiv \mathcal{A}_\theta(C^\infty(\mathbb{R}^2)) = C^\infty(\mathbb{R}^2) \otimes \mathcal{A}_\theta$, i.e., $\phi^i(z, \bar{z}) \to \widehat{\phi}^i(z, \bar{z}) \in \mathcal{A}_\theta^2$. The worldsheet $\mathbb{R}^2$ may be replaced by $\mathbb{T}^2$, $\mathbb{R} \times \mathbb{S}^1$ or $\mathbb{S}^2$. Let us clarify the notation $\mathcal{A}_\theta^2$ after the Wick rotation of the worldsheet coordinate $\tau = it$, so $\mathbb{R}^2 \to \mathbb{R}^{1,1}$. Consider a generic element in the NC $\star$-algebra $\mathcal{A}_\theta^2$ given by

$$\widehat{f}(t, \sigma, y) \in \mathcal{A}_\theta^2. \tag{A74}$$

The matrix representation (41) is now generalized to

$$\widehat{f}(t, \sigma, y) = \sum_{n,m=1}^{\infty} |n\rangle\langle n|\widehat{f}(t, \sigma, y)|m\rangle\langle m| = \sum_{n,m=1}^{\infty} f_{nm}(t, \sigma)|n\rangle\langle m| \tag{A75}$$

where the coefficients $f_{nm}(t, \sigma) := [f(t, \sigma)]_{nm}$ are elements of a matrix $f(t, \sigma)$ in $\mathcal{A}_N^2 \equiv \mathcal{A}_N(C^\infty(\mathbb{R}^{1,1})) = C^\infty(\mathbb{R}^{1,1}) \otimes \mathcal{A}_N$ as a representation of the observable (A74) on the Hilbert space (17). Then, we have an obvious generalization of the duality chain (47) as follows:

$$\mathcal{A}_N^2 \implies \mathcal{A}_\theta^2 \implies \mathfrak{D}^2. \tag{A76}$$

The module of derivations is similarly a direct sum of the submodules of horizontal and inner derivations [67]:

$$\mathfrak{D}^2 = \text{Hor}(\mathcal{A}_N^2) \oplus \mathfrak{D}(\mathcal{A}_N^2) \cong \text{Hor}(\mathcal{A}_\theta^2) \oplus \mathfrak{D}(\mathcal{A}_\theta^2), \tag{A77}$$

where horizontal derivations are locally generated by a vector field

$$k(t, \sigma, y)\frac{\partial}{\partial t} + l(t, \sigma, y)\frac{\partial}{\partial \sigma} \in \text{Hor}(\mathcal{A}_\theta^2). \tag{A78}$$

It can be shown [26,27] that the matrix model for the duality chain (A76) is given by

$$S = -\frac{1}{g_s^2} \int d^2\sigma Tr\left(\frac{1}{4}F_{\alpha\beta}^2 + \frac{1}{2}(D_\alpha\phi_a)^2 - \frac{1}{4}[\phi_a, \phi_b]^2\right), \tag{A79}$$

where $a = 2, \cdots, 2n+1$ and $\sigma^\alpha = (t, \sigma)$, $\alpha = 0, 1$ and $F_{\alpha\beta} = \partial_\alpha A_\beta - \partial_\beta A_\alpha - i[A_\alpha, A_\beta]$. The $n = 4$ case is known as the matrix string theory that is supposed to describe a nonperturbative type IIA string theory in light-cone gauge [49]. The matrix string theory can also be obtained from the BFSS matrix model via compactification on a circle [58]. To achieve this model, the BFSS matrix model has to have nine adjoint scalar fields, $\phi_a(t)$ $(a = 1, \cdots, 9)$, unlike the action (68) with an even number of adjoint scalar fields. The equivalence (75) can be realized only in the case of an even number of adjoint scalar fields. In this case, the action (68) can be understood as a Hilbert space representation of a certain NC gauge theory

under a symplectic vacuum such as (10) with $\text{rank}(B) = 2n$. However, we do not know of a corresponding NC gauge theory where Hilbert space representation precisely reproduces the BFSS matrix model. Fortunately, the matrix string theory (A79) has eight adjoint scalar fields for $n = 4$. Thus, it is possible to realize it as the Hilbert space representation of $(9 + 1)$-dimensional NC $U(1)$ gauge theory with $\text{rank}(B) = 8$.

It will be interesting to understand how to derive the matrix string theory (A79) from the MQM (68) as if the latter has been derived from a contact structure of the zero-dimensional matrix model (6). The basic idea is similar to the scheme to construct the one-dimensional matrix model (68) through the contact structure of zero-dimensional matrices. A difference is that we start with the one-dimensional matrix model (68) and introduce an additional contact structure along a spatial direction whose coordinate is called $\sigma$ in our case. Ultimately, the matrix string theory (A79) can be realized as the quantization of a regular 2-contact manifold. See Ref. [74] for a general $k$-contact manifold. First, let us consider the projection $\pi_2 : \mathbb{R}^{1,1} \times M \to M$, $\pi_2(\sigma^\alpha, x) = x$ where $M$ is a symplectic manifold with the symplectic form $B$.[17] The regular 2-contact $(2n + 2)$-dimensional manifold is defined by a quartet $(\mathbb{R}^{1,1} \times M, \widetilde{B}, \eta^\alpha)$, $\alpha = 0, 1$, where $\widetilde{B} = \pi_2^* B$, such that

$$\eta^0 \wedge \eta^1 \wedge B^n \neq 0 \tag{A80}$$

everywhere and $d\eta^\alpha = \gamma^\alpha B$ with constants $\gamma^\alpha$ and $dB = 0$. Moreover, there are uniquely defined two Reeb vectors $R_\alpha$ $(\alpha = 0, 1)$ satisfying

$$\iota_{R_\alpha} \eta^\beta = \delta_\alpha^\beta, \qquad \iota_{R_\alpha} B = 0, \qquad \alpha, \beta = 0, 1. \tag{A81}$$

The above relations imply

$$\mathcal{L}_{R_\alpha} \eta^\beta = 0, \qquad \mathcal{L}_{R_\alpha} B = 0, \qquad [R_0, R_1] = 0. \tag{A82}$$

For example, the contact forms for the matrix string theory (A79) are given by

$$\eta^0 = dt - \frac{1}{2} p_a dy^a, \qquad \eta^1 = d\sigma - \frac{1}{2} p_a dy^a, \tag{A83}$$

which determines the corresponding Reeb vectors

$$R_0 = \frac{\partial}{\partial t}, \qquad R_1 = \frac{\partial}{\partial \sigma}. \tag{A84}$$

These Reeb vectors span the space of horizontal derivations in Equation (A78).

Since there are two independent contact structures, each contact structure generates its own Hamiltonian vector field defined by (A42). For the contact structures in Equation (A83), they are given by

$$V_\alpha = \frac{\partial}{\partial \sigma^\alpha} + A_\alpha^\mu(t, \sigma, y) \frac{\partial}{\partial y^\mu}. \tag{A85}$$

The quantization of the 2-contact manifold $(\mathbb{R}^{1,1} \times M, \widetilde{B}, \eta^\alpha)$ is simple because it is performed using the Darboux coordinates $(\sigma^\alpha, y^a)$. It is basically defined by the quantization of the symplectic manifold $(M, B)$ in which $\sigma^\alpha$ are regarded as classical variables like the time coordinate in the algebra $\mathcal{A}_\theta^1$. After quantization, a generic element of the NC $\star$-algebra $\mathcal{A}_\theta^2$ takes the form (A74). Then, the module $\mathfrak{D}^2$ in Equation (A77) is generated by

$$\mathfrak{D}^2 = \left\{ \widehat{V}_A(t, \sigma) = \left( \widehat{V}_\alpha, \widehat{V}_a \right)(t, \sigma) | \widehat{V}_\alpha(t, \sigma) = \frac{\partial}{\partial \sigma^\alpha} + \text{ad}_{\widehat{A}_\alpha}, \ \widehat{V}_a(t, \sigma) = \text{ad}_{\widehat{\varphi}_a} \right\}, \tag{A86}$$

where $A = 0, 1, \cdots, 2n + 1$ and the adjoint operations are inner derivations of $\mathcal{A}_\theta^2$. In the commutative limit, the module (A86) reduces to ordinary vector fields $V_A = (V_\alpha, V_a) \in$

$\mathfrak{X}(\mathcal{M})$ and it is related to the orthonormal frames by $V_A = \lambda E_A$. Finally, the corresponding Lorentzian metric dual to the matrix string theory (A79) is given by [26,27]

$$ds^2 = \lambda^2 \eta_{AB} v^A \otimes v^B = \lambda^2 \left( \eta_{\alpha\beta} d\sigma^\alpha d\sigma^\beta + v_\mu^a v_\nu^a (dy^\mu - \mathbf{A}^\mu)(dy^\nu - \mathbf{A}^\nu) \right), \tag{A87}$$

where $\mathbf{A}^\mu := A_\alpha^\mu(t, \sigma, y) d\sigma^\alpha$ and $\lambda^2 = v_{(t,\sigma)}(V_0, V_1, \cdots, V_{2n+1})$ is determined by the volume-preserving condition, $\mathcal{L}_{V_A} v_{(t,\sigma)} = 0$, with respect to a given volume form

$$v_{(t,\sigma)} = dt \wedge d\sigma \wedge v = \lambda^2 dt \wedge d\sigma \wedge v^1 \wedge \cdots \wedge v^{2n}. \tag{A88}$$

Instead of the conformal frame $V_A = \lambda E_A$, one may choose another frame, the so-called comoving frame, similar to Equation (57):

$$V_A = (V_\alpha, V_a) = (E_\alpha, \lambda E_a). \tag{A89}$$

The $(2n + 2)$-dimensional Lorentzian metric is then given by

$$ds^2 = \eta_{AB} e^A \otimes e^B = \eta_{\alpha\beta} d\sigma^\alpha d\sigma^\beta + \lambda^2 v_\mu^a v_\nu^a (dy^\mu - \mathbf{A}^\mu)(dy^\nu - \mathbf{A}^\nu). \tag{A90}$$

This comoving frame may be more convenient to incorporate the inflation metric (130).

Let us come back to our previous question about the generalization of pseudoholomorphic curves. In order to address this issue, let us consider the Wick rotation $t = -i\tau$ again to return to the Euclidean space. If the quantum version of pseudoholomorphic curves exists, Equation (A61) suggests that it will also obey the first-order partial differential equations. It is well known [100,101] that the matrix string action (A79) admits such a first-order system. For simplicity, assume that adjoint scalar fields mostly vanish except $(\phi_2, \phi_3) \neq 0$. It is convenient to use the complex variables

$$\phi = \frac{1}{2}(\phi_2 - i\phi_3), \qquad \phi^\dagger = \frac{1}{2}(\phi_2 + i\phi_3). \tag{A91}$$

It is not difficult to show that the Euclidean action with $\phi_a = 0$ for $a = 4, \cdots, 9$ can be written as the Bogomol'nyi type, i.e.,

$$\begin{aligned} S &= \frac{1}{g_s^2} \int d^2\sigma \, Tr \left( \frac{1}{4} F_{\alpha\beta}^2 + \frac{1}{2}(D_\alpha \phi_a)^2 - \frac{1}{4}[\phi_a, \phi_b]^2 \right) \\ &= \frac{2}{g_s^2} \int d^2\sigma \, Tr \left( \left( iF_{z\bar{z}} - [\phi, \phi^\dagger] \right)^2 + |D_{\bar{z}}\phi|^2 - i\partial_\alpha \left( \varepsilon^{\alpha\beta} \phi^\dagger D_\beta \phi \right) \right). \end{aligned} \tag{A92}$$

Since the last term is a topological number, the minimum of the action is achieved in the configurations obeying

$$F_{z\bar{z}} + i[\phi, \phi^\dagger] = 0, \qquad D_{\bar{z}}\phi = 0. \tag{A93}$$

Note that the above equations recover Equation (A65) in a very commutative limit where $[\phi^\dagger, \phi] = 0$. Therefore, it is reasonable to identify Equation (A93) with the quantum version of pseudoholomorphic curves.

Mathematically, Equation (A93) is equivalent to the Hitchin equations describing a Higgs bundle [51,52]. A Higgs bundle is a system composed of a connection $A$ on a principal $G$-bundle or simply a vector bundle $E$ over a Riemann surface $\Sigma$ and a holomorphic endomorphism $\phi$ of $E$ satisfying Equation (A93). The Hitchin equations describe four-dimensional Yang–Mills instantons on $\Sigma \times \mathbb{R}^2$, which are invariant with respect to the translation group $\mathbb{R}^2$. (This $\mathbb{R}^2$ is transverse to the Riemann surface.) Using the translation invariance, the Yang–Mills instantons can be dimensionally reduced to the Riemann surface $\Sigma$ in which Yang–Mills gauge fields along the isometry directions become an adjoint Higgs field $\phi$. In our case, the gauge group $G$ is $U(N)$. In particular, we are interested in the large $N$ limit, i.e., $N \to \infty$. In this limit, the action (A92) can be mapped to four-dimensional NC $U(1)$ gauge theory under the Coulomb branch vacuum $\langle \phi_a \rangle_{\text{vac}} = p_a$, $a = 2, 3$ obeying

the commutation relation $[p_2, p_3] = -iB_{23}$. Then, the Hitchin equations (A93) precisely become the self-duality equation for NC $U(1)$ instantons on $\Sigma$ (or $\mathbb{R}^2$) $\times \mathbb{R}_\theta^2$ [102–104]. The corresponding gravitational metric for the case $n = 1$ can be identified with Equation (A87) with the analytic continuation $t = -i\tau$. It was shown in [57,84,105] that the solution of the Hitchin equations (A93) is dual to four-dimensional gravitational instantons which are hyperKähler manifolds. In particular, the real heaven is governed by the $su(\infty)$ Toda equation and the self-duality equation for the real heaven exactly reduces to the commutative limit of the Hitchin equations (A93). See Equation (4.31) in Ref. [84]. Thus, the Hitchin system with the gauge group $G = U(N \to \infty)$ may be closely related to the Toda field theory. Indeed, this interesting connection was already analyzed in [106]. In sum, Hitchin's equations, NC $U(1)$ instantons, gravitational instantons and pseudoholomorphic curves may be only the tip of the iceberg in matrix string theory (A79) that have barely shown themselves.

Let us conclude this section by drawing an invaluable insight. We have observed that NC spacetime is much more radical and mysterious than we thought before. It is fair to say that we have not yet fully understood the mathematical foundation of NC spacetime. A remarkable point is that NC spacetime necessarily implies emergent spacetime if spacetime at microscopic scales should be viewed as NC. This means that classical spacetime is a derived concept from something deeper. A pseudoholomorphic curve is a stringy generalization of a geodesic worldline in Riemannian geometry [50]. Recall that the pseudoholomorphic curve is basically a minimal surface or a string worldsheet embedded into spacetime. However, to make sense of the emergent spacetime picture, we need a mathematically precise framework for describing strings in a background-independent way. The background-independent theory must give up the picture that strings are vibrating in a preexisting spacetime. In this Appendix, we have aimed at clarifying how the pseudoholomorphic curves can be lifted to aN NC spacetime by the matrix string theory. The matrix string theory naturally extends the first-quantized string theory so that it also describes the nonperturbative interactions of the splitting and joining of strings, producing surfaces with nontrivial topology [49]. That is, the matrix string theory is a second-quantized theory in which spacetime emerges from the collective behavior of matrix strings. Thus, we argue that the NC spacetime can be viewed as a second-quantized string from the perspective of the background-independent formulation of quantum gravity.

## Notes

1    George Santayana (1863–1952).

2    Graham Ross in *Quanta magazine* "At multiverse impasse, a new theory of scale" (18 August 2014) and *Wired.com* "Radical new theory could kill the multiverse hypothesis".

3    Even though our work was motivated by Ref. [6], it should be pointed out that the conclusions in [6] can be avoided in the so-called emergent universe scenario [7–9] and some models were also presented to cure the instabilities of the emergent universe [10,11].

4    Nonetheless, the friction term does not lead to dissipative energy production. This fact can be seen by observing that Equation (3) can be derived from the first law of thermodynamics, $dE + pdV = Vd\rho + (\rho + p)dV = 0$, where $\rho + p = \dot\phi^2$ and $\dot\rho = \left( \ddot\phi + \frac{\delta V}{\delta\phi} \right)\dot\phi$.

5    Here we refer to the background-independent theory in which any spacetime structure is not a priori assumed but defined by the theory.

6    The conventional choice of vacuum in Coulomb branch is given by $[\phi_a, \phi_b]|_{\text{vac}} = 0$ and so $\langle\phi_a\rangle_{\text{vac}} = \text{diag}\big((\lambda_a)_1, (\lambda_a)_2, \cdots, (\lambda_a)_N\big)$. However, it turns out (see Section III.C in [26]) that, in order to describe a classical geometry from a background-independent theory, it is necessary to have a nontrivial vacuum defined by a "coherent" condensation obeying the algebra (10). For this reason, we will choose the Moyal–Heisenberg vacuum instead of the conventional vacuum. A similar reasoning was also advocated in footnote 2 in Ref. [54].

7    We will often use the symbol $M$ to denote a generic manifold whereas the symbol $\mathcal{M}$ has been used to emphasize an emergent manifold.

8    However, we point out that there is another approach for the emergent time where time is regarded as a dynamical variable, for example, Ref. [43]. Therefore, our approach for the emergent time must be considered as an alternative viewpoint.

9    Note that the vacuum solution (9) is further degenerated under the scaling $p_a \to p_a' = \beta p_a$ or $y^a \to y'^a = \beta^{-1} y^a$ as far as $\beta \in \mathbb{R} \setminus \{0\}$ is a nonzero constant. We will use this freedom to normalize the initial length scale such that $|y^a(t = 0)| = L_P$ or $l_s = \sqrt{\alpha'}$.

10　This experiment is a simple twist of the well-known solution of Gauss's law for gravity inside the earth, in which the minus sign in the gravitational potential energy presupposes a repulsive force rather than the usual attractive force. The repulsive force in Newtonian gravity is given by $\mathbf{F} = k_g \mathbf{r} = -\nabla V(r)$ where $k_g = \frac{4\pi G_N m \rho_{\mathrm{vac}}}{3}$ and $V(r) = -\frac{G_N M(r) m}{2r}$ is the gravitational potential energy. Note that the change of sign and the factor 2 enhancement are due to the general relativity effect since $\frac{\ddot{a}}{a} = -\frac{4\pi G_N}{3}(\rho_{\mathrm{vac}} + 3p) = -\frac{4\pi G_N}{3}(-2\rho_{\mathrm{vac}})$.

11　Note that $a = b + d\lambda$ where $b = -p_i dq^i$ and $\lambda = \frac{1}{2} q^i p_i$. Thus one can also define the conformal vector field $X$ by $\iota_X \omega = \kappa b + dH'$ where $H' = H + \kappa\lambda$. In this case $X = \kappa p_i \frac{\partial}{\partial p_i} + X_{H'}$ and the equations of motion are given by $\frac{dq^i}{dt} = \frac{\partial H'}{\partial p_i}$ and $\frac{dp_i}{dt} = \kappa p_i - \frac{\partial H'}{\partial q^i}$. For $H' = \frac{1}{2}(p_i^2 + \omega^2 q_i^2)$, the general solution is $q^i(t) = A^i e^{\frac{\kappa}{2}t} \sin\left(\sqrt{\omega^2 - \frac{\kappa^2}{4}}\, t + \theta\right)$, which describes a damped harmonic oscillator when $\kappa < 0$. However the vector field defined by Equation (88) is more convenient for our purpose.

12　One important consequence is that the energy will not be positive. Polyakov has suggested [78,79] that this makes de Sitter space unstable with respect to decay by the creation of particle–antiparticle pairs.

13　This feature is due to our choice of a coordinate frame to describe the dynamical system. The time evolution operator $\widehat{\phi}_0(t, y) = i\frac{\partial}{\partial t} + \widehat{A}_0(t, y)$ is defined in the comoving frame. In general, one may choose an arbitrary frame in which the time evolution is described by $k(t, y)\frac{\partial}{\partial t} \in \mathrm{Hor}(\mathcal{A}_\theta^1)$. A particularly interesting frame is the conformal coordinates with which the metric is given by $ds^2 = a(\eta)^2(-d\eta^2 + d\mathbf{x} \cdot d\mathbf{x})$ where $a(\eta) = -\frac{1}{H\eta}$ and $-\infty < \eta < 0$. The conformal coordinates can be easily transformed to the comoving coordinates by $a(\eta)d\eta = dt$.

14　It may be remarked that it is not possible to realize the Liouville vector field in terms of a local Hamiltonian function. Thus, the inflation is a dynamical system without Hamiltonian. However, we present some examples in Appendix B showing that this situation may be cured by introducing a time-dependent Hamiltonian.

15　There is a nice exposition on the Landau damping by Werner Herr, "Introduction to Landau Damping", available at: https://cds.cern.ch/record/1507631/files/CERN-2014-009.pdf (accessed on 21 December 2023). Recently the Landau damping has even been mathematically established at the nonlinear level [91].

16　In the topological A-model that is concerned with pseudoholomorphic maps from $\Sigma$ to $M = T^*N$, there is a vanishing theorem [99] stating that $\int_\Sigma f^*B = 0$. In particular, the mappings from $\partial\Sigma$ to $N$ are necessarily constant.

17　It is possible to replace $\mathbb{R}^{1,1} \times M$ by a general $(2n + 2)$-dimensional manifold $N$ as far as there is a well-defined two-dimensional foliation $\mathcal{V}$ such that the corresponding space of leaves $N/\mathcal{V} = M$ is a Hausdorff differentiable manifold [74]. See (A24) for a relevant discussion. We will keep the maximal simplicity for a plain argument.

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
