# Peer review of "Emergent Spacetime and Cosmic Inflation"

_universe, doi:10.3390/universe10030150_

Round 1
Reviewer 1 Report
Comments and Suggestions for Authors
The paper contains novel ideas on important topic but is not accessible to a lay theorist given its specialised knowledge.
Reviewer 2 Report
Comments and Suggestions for Authors
Comments are in the attached file

Reviewer 3 Report
Comments and Suggestions for Authors
This paper is a valuable contribution to the subject of the origin of space-time and cosmic inflation and moreover eliminates the need of a multiverse world. So, I consider it must be accepted for publication. However, it would be beneficial for the readers if the author would elaborate a bit on how fits the origin of cosmic structure in this scenario.
Reviewer 4 Report
Comments and Suggestions for Authors
I think the paper is nice and describes the possibility of an Emergent space time whose structure could be crucial for the understanding of the Early Universe. However it is strongly motivated by reference 6 of this paper, A. Borde, A. H. Guth and A. Vilenkin, Inflationary spacetimes are incomplete in past directions, Phys. Rev. Lett. 90 (2003) 151301, 1446 [gr-qc/0110012]. The conclusions of this paper can be avoided in the so called Emergen Universe scenarios, see for example, The emergent universe: Inflationary cosmology with no singularity
- George F.R. Ellis
- Roy Maartens
-
Published in:
- Class.Quant.Grav. 21 (2004) 223-232
-
e-Print:
- gr-qc/0211082 [gr-qc], this has been followed by other models that cure instabilities of this model , for example,
in
Emergent Cosmology, Inflation and Dark Energy- Eduardo Guendelman
- Ramón Herrera
- Pedro Labrana ,
- Emil Nissimov
-
Svetlana Pacheva
-
Published in:
- Gen.Rel.Grav. 47 (2015) 2, 10
-
e-Print:
- 1408.5344 [gr-qc]
so these papers invalidate (by finding counter examples) to the claims in ref. 6.
So the author can still continue with all his development, can cite ref. 6, but then should cite the papers above and others he may find that avoid the findings in 6 and do all the development he does. After this the paper can be published.
Round 2
Reviewer 2 Report
Comments and Suggestions for Authors
The file with review is attached
